# Beyond Loss Guidance: Using PDE Residuals as Spectral Attention in Diffusion Neural Operators

## Abstract

Diffusion-based solvers for partial differential equations (PDEs) are often bottlenecked by slow gradient-based test-time optimization routines that use PDE residuals for loss guidance. They additionally suffer from optimization instabilities and are unable to dynamically adapt their inference scheme in the presence of noisy PDE residuals. To address these limitations, we introduce PRISMA (PDE Residual Informed Spectral Modulation with Attention), a conditional diffusion neural operator that embeds PDE residuals directly into the model's architecture via attention mechanisms in the spectral domain, enabling gradient-descent free inference. In contrast to previous methods that use PDE loss solely as external optimization targets, PRISMA integrates PDE residuals as integral architectural features, making it inherently fast, robust, accurate, and free from sensitive hyperparameter tuning. We show that PRISMA has competitive accuracy, at substantially lower inference costs, compared to previous methods across five benchmark PDEs especially with noisy observations, while using 10x to 100x fewer denoising steps, leading to 15x to 250x faster inference.

## 1 Introduction

Given the ubiquitous presence of partial differential equations (PDEs) in almost every scientific discipline, there is a rapidly growing literature on using neural networks for solving PDEs (Raissi et al., 2019a; Lu et al., 2019). This includes seminal works in *operator learning* methods such as the Fourier Neural Operator (FNO) Li et al. (2020) that learns resolution-independent mappings between function spaces of input parameters $\mathbf{a}$ and solution fields $\mathbf{u}$. However, a major limitation of these methods is their reliance on complete and clean observations of either $\mathbf{a}$ or $\mathbf{u}$, a condition rarely met in real-world applications where data is inherently noisy and sparse.

The rise of *generative models* has inspired another class of methods for solving PDEs by modeling the joint distribution of $\mathbf{a}$ and $\mathbf{u}$ using diffusion-based backbones (Huang et al., 2024; Yao et al., 2025; Lim et al., 2023; Shu et al., 2023; Bastek et al., 2024; Jacobsen et al., 2025). These methods offer two key advantages over operator learning methods: (i) they generate full posterior distributions of $\mathbf{a}$ and/or $\mathbf{u}$, enabling principled uncertainty quantification crucial for ill-posed inverse problems, and (ii) they naturally accommodate sparse observations during inference using likelihood-based and PDE residual-based *loss guidance*, termed diffusion posterior sampling or test-time optimization.

Despite these advances, current diffusion-based PDE solvers suffer from three major limitations. *First*, they are inherently slow during inference due to the large number of denoising steps required by their *unconditional training* paradigm, which is further burdened by expensive test-time optimization when PDE-residual loss is optimized. Second, the *minimization of PDE residuals as external loss* introduces optimization instabilities due to the sensitive nature of PDE loss gradients at different stages of inference, requiring careful hyperparameter tuning (Zhang & Zou, 2025; Cheng et al., 2024; Utkarsh et al., 2025). *Third*, when observations are corrupted by unknown noise structures (e.g., sensor failures or environmental interference), these methods cannot *dynamically adapt* their guided inference scheme to discount noisy PDE residuals, resulting in degraded performance.

At the heart of these limitations lies a fundamental bottleneck in existing generative frameworks for solving PDEs: "treating PDE constraints as *external optimization targets* rather than as *integral*

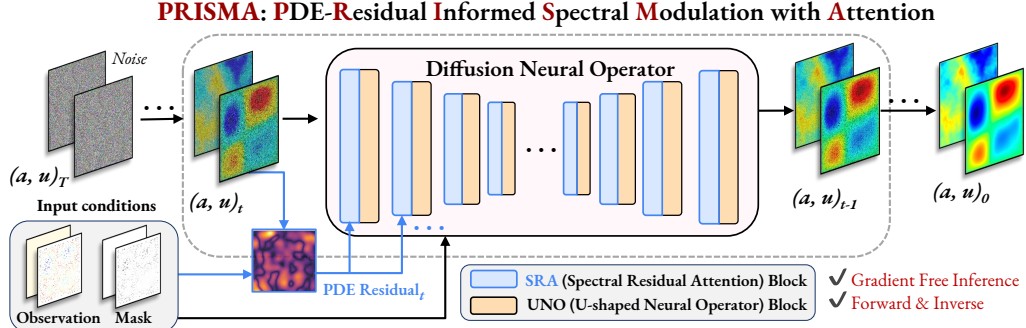

**PRISMA**: PDE-**R**esidual **I**nformed **S**pectral **M**odulation with **A**ttention

Figure 1: The **PRISMA** inference pipeline. We use a U-shaped Diffusion Neural Operator (UNO) to iteratively refine estimates of $\mathbf{a}$ and $\mathbf{u}$ starting from noise $(\mathbf{a}, \mathbf{u})_T$ to a clean solution $(\mathbf{a}, \mathbf{u})_0$. At each denoising step, PDE residuals are architecturally injected via a novel SRA block at every layer of the UNO, enabling fast, gradient-descent free inference for both forward and inverse problems.

*architectural features*." Since previous methods employ PDE residuals solely as additional loss terms during training or inference, they face a challenging multi-objective optimization problem that is hard to stabilize and requires careful hyper-parameter tuning (Wang et al., 2021; Farea et al., 2025). Moreover, by reducing spatially-resolved PDE residual fields to scalar loss terms, existing methods discard the rich *point-wise information* that could otherwise provide local guidance for fine-grained steering. In contrast to using PDE residuals as loss guidance, we ask: "is it possible to incorporate the knowledge of PDE residuals directly in the architecture of diffusion-based models, avoiding slow and sensitive loss gradients during inference altogether?"

We introduce **PRISMA** (PDE Residual Informed Spectral Modulation with Attention), a *conditional* diffusion neural operator that embeds PDE residuals into the architecture via *attention mechanisms* instead of loss guidance, enabling gradient-descent free inference. A key innovation in PRISMA is a novel **Spectral Residual Attention (SRA)** block (Figure 1) that is injected at every layer of a U-shaped denoiser backbone to locally attend to the model's generated outputs based on guidance from the PDE residual field, during both training and inference. By integrating PDE residuals as architectural features directly in the *spectral domain*, SRA is able to modulate and *dynamically adapt* to the local contributions of varying frequency modes in the PDE residual space, even in the presence of noise. Another innovation in PRISMA is the use of input masks in the architecture to regulate the generation of $\mathbf{a}$ and $\mathbf{u}$ under varying conditions of input observations, rather than training an unconditional model guided only at inference. This conditional design enables PRISMA to serve as a unified framework for solving PDEs: a single versatile model capable of addressing both forward and inverse problems under full, sparse, or noisy observations, without task-specific inference.

Our work makes four key contributions. *(1)* We introduce a novel strategy for incorporating physical knowledge in diffusion models as **architectural guidance**, where the PDE residual is embedded directly as an internal feature in the model instead of a test-time guidance loss. *(2)* A powerful outcome of this design is **gradient-descent free inference**, which fundamentally removes the need for costly and sensitive optimization of likelihood and PDE loss terms during inference, forming the basis of our method's speed and robustness. *(3)* We show **15x to 250x faster inference (time)** compared to baselines across a diverse set of benchmark PDEs with varying problem configurations (forward and inverse), while achieving superior accuracy especially in the presence of noisy observations. *(4)* Our work also represents the first *conditional design* of a generative model trained to **unify forward and inverse problems** under varying configurations of input observations.

## 2 RELATED WORKS

**Neural PDE solvers:** Deep learning methods such as Physics-Informed Neural Networks (PINNs) (Raissi et al., 2019b) offer a powerful alternative to classical methods for solving PDEs by minimizing PDE residual loss during training. However, PINNs are prone to optimization and scaling

Table 1: Comparing PRISMA with previous methods for solving PDEs in terms of their strategy for integrating PDE residuals, their ability to solve forward and inverse problems using a single unified model, and their adaptability to the presence of noise in PDE residuals. PRISMA is unique from other works in terms of the use of PDE residuals in the spectral domain, with point-wise architectural guidance that is adaptive to noise, resulting in gradient-descent free inference. We also compare the forward error and inference time per sample for the Darcy equation.

| Method | PDE Residual Integration Strategy | | | | Unified Model | Adaptive to Noise | Inf. Time / Sample (s)↓ | Forward (%) Error↓ |
|---|---|---|---|---|---|---|---|---|
| | Integration Domain | Point-wise Guidance | Architectural Guidance | Gradient-Descent Free Inference | | | | |
| PINN | Spatial | × | × | - | × | × | 3.3 | 27.3 |
| FNO | - | × | × | - | × | × | 0.1 | 2.3 |
| DeepONet | - | × | × | - | × | × | 0.09 | 25.6 |
| PINO | Spatial | × | × | - | × | × | 0.11 | 1.1 |
| DiffusionPDE | Spatial | × | × | × | ✓ | × | 213.0 | 2.4 |
| FunDPS | Spatial | × | × | × | ✓ | × | 11.8 | 1.6 |
| **PRISMA** | **Spectral** | ✓ | ✓ | ✓ | ✓ | ✓ | **0.18** | **0.94** |

challenges (Zhongkai et al., 2024; Liu et al., 2024a), motivating operator-learning methods such as FNO and DeepONet, which learn resolution-invariant mappings between functions of input parameters and solution fields for fast inference (Li et al., 2020; Rahman et al., 2022; Kovachki et al., 2023; Lu et al., 2019). To further enhance physics guidance in FNO architecture, PINO (Li et al., 2024) incorporates physics-based loss during training and inference optimization. However, similar to PINNs, PINO suffers from the same optimization issues due to the PDE loss-based training. Another limitation with PINNs and operator learning methods is that they produce *deterministic* estimates for either the forward or inverse problem and typically assume clean, dense data, offering no native uncertainty.

**Generative models for PDEs in Function Spaces:** Diffusion models enable sampling from distributions over fields, providing uncertainty quantification and flexible conditioning. For example, *DiffusionPDE* Huang et al. (2024) learns the joint probability distribution of $(\mathbf{a}, \mathbf{u})$ using a diffusion model during training, and employs PDE-residual guidance as a loss term during inference to produce physically consistent solutions. By minimizing PDE residuals during inference, DiffusionPDE can work with arbitrary forms of sparsity in $\mathbf{a}$ and/or $\mathbf{u}$ that has not been seen during training. However, since DiffusionPDE operates directly in the native spatial domain, it remains tied to a *fixed spatial resolution* of $\mathbf{a}$ and $\mathbf{u}$. Newer methods expand the design space by incorporating score-based or diffusion backbones for physics (Shu et al., 2023; Bastek et al., 2024; Jacobsen et al., 2025; Li et al., 2025). Because PDE states are functions, not images, several works lift generative modeling to function spaces for *resolution independence*: Denoising Diffusion Operators (DDOs) perturb Gaussian random fields and provide discretization robustness (Lim et al., 2023; Yao et al., 2025); and infinite-dimensional diffusion frameworks formalize well-posedness and dimension-free properties (Pidstrigach et al., 2023). Similarly, a recent line of work on *FunDPS* (Function-space Diffusion Posterior Sampling) Yao et al. (2025) uses U-shaped neural operators (UNO) Rahman et al. (2022) as the denoising backbone in diffusion models, leading to a new framework of *diffusion neural operators*. By marrying the strengths of neural operators and diffusion models, FunDPS can generate posterior distributions of $\mathbf{a}$ or $\mathbf{u}$ while being resolution-agnostic.

**Limitations of Diffusion-based PDE solvers:** Existing diffusion-based methods mostly learn an unconditional distribution of $(\mathbf{a}, \mathbf{u})$ during training, and apply physics as external, spatial loss aggregating residuals rather than using their full point-wise structure. (Zhang & Zou, 2025; Cheng et al., 2024; Utkarsh et al., 2025). As a result, unconditional diffusion models with DPS-style guidance require multiple denoising steps and are sensitive to guidance hyper-parameter tuning (Chung et al., 2022; Huang et al., 2024). Training-time alternatives reduce this burden, e.g., conditioning on measurements (Shysheya et al., 2024; Zhang & Zou, 2025), yet still implement physics as an *external objective*. Moreover, optimizing PDE residuals during sampling introduces known instabilities and tuning overhead (Zhang & Zou, 2025; Cheng et al., 2024; Utkarsh et al., 2025).

**Loss guidance vs. Architecture guidance:** Table 1 summarizes how PRISMA is different from the baselines, specifically in the way it uses PDE residual as an architectural feature. Our work is related to the growing literature on enforcing constraints via architectural integration to provide stronger,

more stable physics than soft penalties (Trask et al., 2022; Liu et al., 2024b). While shallow variants concatenate physics features to inputs (Shu et al., 2023; Takamoto et al., 2023), deeply integrating the PDE residual as a learnable architectural feature, especially in spectral domains where neural operators capture global behavior, remains largely unexplored, particularly for diffusion models.

## 3 PRISMA: PROPOSED UNIFIED FRAMEWORK FOR SOLVING PDEs

We develop PRISMA as an all-purpose generative framework to model the joint distribution of PDE parameters and solutions conditioned on varying configurations of input observations. PRISMA is designed to be highly flexible for conditional generation of outputs in both forward and inverse problems, with complete, partial, or even noisy observations. In the following, we first describe the unified problem statement of PRISMA and provide a background on diffusion neural operators that we use as a backbone in our framework, before introducing the PRISMA model architecture.

### 3.1 PROBLEM STATEMENT FOR UNIFIED GENERATIVE MODELING OF PDEs

Given a spatial domain $\Omega \subset \mathbb{R}^d$, we are interested in solving parametric PDEs of the form

$$\mathcal{N}(\mathbf{u}(c); \mathbf{a}(c)) = 0, \quad c \in \Omega, \tag{1}$$

subject to some initial and boundary conditions where $\mathbf{a} \in \mathcal{A}$ represents the parameter field of the PDE (e.g., material coefficients or source terms), $\mathbf{u} \in \mathcal{U}$ denotes the solution field, and $\mathcal{N}$ is a non-linear differential operator. There are two main classes of problems when solving PDEs: (i) the *forward problem*, where we want to generate $\mathbf{u}$ given observations of parameters, $\mathbf{a}_{\text{obs}}$, and the *inverse problem*, where we want to generate $\mathbf{a}$ given observations of $\mathbf{u}_{\text{obs}}$. Optionally, we may have sparsity or noise in $\mathbf{a}_{\text{obs}}$, $\mathbf{u}_{\text{obs}}$, or both, arising from incomplete or inaccurate observations.

We consider a *unified framework* for solving both forward and inverse problems by training a single model to learn the joint probability of $(\mathbf{a}, \mathbf{u})$ conditioned on any partial set of observations, $(\mathbf{a}_{\text{obs}}, \mathbf{u}_{\text{obs}})$. To encode varying configurations of input observations in the conditioning of the joint model, we introduce binary masks $(\mathbf{M}_a, \mathbf{M}_u)$ that serve two uses: (i) they indicate the sparsity patterns in $\mathbf{a}_{\text{obs}}$ and $\mathbf{u}_{\text{obs}}$, and (ii) they specify whether we are solving the forward or inverse problem. A value of 1 in these masks indicates that an observation is present, while 0 indicates no observation. Table 2 summarizes the different configurations of input conditions that result in different problem settings, unifying forward and inverse problems as well as full and sparse observations.

Table 2: Characterizing varying configurations of input conditions in our framework for solving different settings of PDE problems. A value of True (False) in masks indicate a tensor of all 1s (0s).

| Problem Setting | Obs. ($\mathbf{a}_{\text{obs}}$) | Obs. ($\mathbf{u}_{\text{obs}}$) | Mask ($\mathbf{M}_a$) | Mask ($\mathbf{M}_u$) |
|---|---|---|---|---|
| Full (Forward) | Full | – | True | False |
| Full (Inverse) | – | Full | False | True |
| Sparse (Forward) | Sparse | – | Sparsity Mask | False |
| Sparse (Inverse) | – | Sparse | False | Sparsity Mask |
| Sparse (Both) | Sparse | Sparse | Sparsity Mask | Sparsity Mask |

### 3.2 BACKBONE: CONDITIONAL U-SHAPED DIFFUSION NEURAL OPERATORS

We model the joint distribution of the concatenated continuous field, $\mathbf{x} = [\mathbf{a}, \mathbf{u}]$ using a conditional Denoising Diffusion Operator (DDO) (Lim et al., 2023), a recent class of score-based generative models designed for function spaces Kerrigan et al. (2022); Pidstrigach et al. (2023). The forward process in DDO perturbs a clean function $\mathbf{x}_0$ with progressively larger noise, yielding a noisy function $\mathbf{x}_\sigma = \mathbf{x}_0 + \sigma\boldsymbol{\epsilon}$ at noise level $\sigma$. To preserve the functional integrity of the data, the noise $\boldsymbol{\epsilon}$ is sampled from a Gaussian Random Field (GRF), $\boldsymbol{\epsilon} \sim \mathcal{N}(0, I)$.

The corresponding reverse diffusion process is learned by a denoiser network, $D_\theta$, parameterized as a neural operator to predict $\mathbf{x}_0$ from $\mathbf{x}_\sigma$. We specifically instantiate $D_\theta$ as a U-shaped Neural Operator (UNO) (Rahman et al., 2022) comprising of a multi-scale hierarchy of $L$ layers, where the

transformation of features from layer $l$ to layer $l + 1$ is defined as:

$$\mathbf{x}^{l+1} = \underbrace{\mathcal{F}^{-1}\Big(W^l \odot \mathcal{F}(\mathbf{x}^l)\Big)}_{\text{Global Spectral Path}} + \underbrace{\psi^l(\mathbf{x}^l)}_{\text{Local Spatial Path}} , \qquad (2)$$

where the global spectral path applies learnable weights $W^l$ in Fourier space using the Fast Fourier Transform ($\mathcal{F}$) and its inverse ($\mathcal{F}^{-1}$), while the local spatial path is a local residual block $\psi^l$.

We consider a conditional variant of UNO by augmenting the inputs to the denoiser network $D_\theta$ with *conditioning information* $\mathbf{y} = \{\mathbf{x}_{\text{obs}} = (\mathbf{a}_{\text{obs}}, \mathbf{u}_{\text{obs}}), \mathbf{M} = (\mathbf{M}_a, \mathbf{M}_u)\}$. We train $D_\theta$ using the Elucidated Diffusion Model (EDM) (Karras et al., 2022) objective, defined as:

$$\mathcal{L}_{\text{EDM}} = \mathbb{E}_{\mathbf{x}_0, \mathbf{y}, \sigma, \boldsymbol{\epsilon}} \left[ \lambda(\sigma) \left\| D_\theta(\mathbf{x}_{\sigma, \epsilon}, \sigma, \mathbf{y}) - \mathbf{x}_0 \right\|_2^2 \right]. \qquad (3)$$

## 3.3 PRISMA MODEL ARCHITECTURE

There are two key innovations that we introduce in our UNO backbone to arrive at PRISMA. First, we compute PDE residuals of $\mathbf{x}$ at every denoising step informed by observations, $\mathbf{x}_{\text{obs}} = (\mathbf{a}_{\text{obs}}, \mathbf{u}_{\text{obs}})$. Second, we inject the residuals in the attention mechanism of a novel **Spectral Residual Attention** (SRA) block applied at every UNO layer. We describe both these innovations as follows:

**Computing Observation-Informed PDE Residuals:** Given some observations $\mathbf{x}_{\text{obs}}$ and inputs to the denoiser network $\mathbf{x} = (\mathbf{a}, \mathbf{u})$, we want to compute the physical consistency of $\mathbf{x}$ at unobserved locations (i.e., locations that are not part of $\mathbf{x}_{\text{obs}}$. To accomplish this, we first copy the information from $\mathbf{x}_{\text{obs}}$ to $\mathbf{x}$ using masks $\mathbf{M} = (\mathbf{M}_a, \mathbf{M}_u)$, obtaining *mixed fields*: $\mathbf{x}_{\text{mix}} = (\mathbf{a}_{\text{mix}}, \mathbf{u}_{\text{mix}})$ as follows:

$$\mathbf{a}_{\text{mix}} = \mathbf{M}_a \odot \mathbf{a}_{\text{obs}} + (1 - \mathbf{M}_a) \odot \mathbf{a}_\sigma, \qquad \mathbf{u}_{\text{mix}} = \mathbf{M}_u \odot \mathbf{u}_{\text{obs}} + (1 - \mathbf{M}_u) \odot \mathbf{u}_\sigma, \qquad (4)$$

where $\mathbf{x}_\sigma = (\mathbf{a}_\sigma, \mathbf{u}_\sigma)$ is the input to the denoiser during training, which can be replaced with $\mathbf{x}_t = (\mathbf{a}_t, \mathbf{u}_t)$ during inference. We then compute PDE residuals, $\mathbf{r} = \mathcal{F}(\mathbf{a}_{\text{mix}}, \mathbf{u}_{\text{mix}})$ (shown visually in Figure 2(left)). Note that $\mathbf{r}$ is computed once at every denoising step at the model's native resolution and then progressively downsampled to provide a multi-resolution guidance signal at every layer of UNO.

**Injecting PDE residuals in SRA:** We adapt the UNO architecture by introducing a novel SRA block inside the global spectral path of every UNO layer The SRA block at layer $l$ first modulates the input feature maps $\mathbf{x}^l$ using the PDE residual $\mathbf{r}$ to produce an intermediate physics-informed state, $\mathbf{x}^l_{\text{SRA}}$. This state is then passed through the global spectral path of the UNO block at layer $l$, while the original feature map $\mathbf{x}^l$ is passed into the local spatial path as:

$$\mathbf{x}^l_{\text{SRA}} = \text{SRA}(\mathbf{x}^l, \mathbf{r}), \qquad\qquad \mathbf{x}^{l+1} = \mathcal{F}^{-1}\Big(W^l \odot \mathcal{F}(\mathbf{x}^l_{\text{SRA}})\Big) + \psi^l(\mathbf{x}^l). \qquad (5)$$

---

**Algorithm 1** PRISMA Model Training

---

**Require:** Data distribution $\mu$, GRF covariance $\mathbf{C}$, noise distribution $p(\sigma)$, PDE operator $\mathcal{R}$, mask sampling distribution $p(\text{task})$.

1: Initialize model parameters $\theta$.
2: **repeat**
3:      Draw clean state $\mathbf{x}_0 = [\mathbf{a}, \mathbf{u}] \sim \mu$ and noise level $\sigma \sim p(\sigma)$.
4:      Sample task-specific masks $(\mathbf{M}_a, \mathbf{M}_u) \sim p(\text{task})$.          {Sample uncond/ for/inv masks}
5:      Set observations $(\mathbf{a}_{\text{obs}}, \mathbf{u}_{\text{obs}}) \leftarrow (\mathbf{a}, \mathbf{u})$.
6:      Sample GRF noise $\eta \sim \mathcal{N}(0, \sigma^2 \mathbf{C})$.
7:      Construct noisy sample $\mathbf{x}_\sigma \leftarrow \mathbf{x}_0 + \eta$. Let $\mathbf{x}_\sigma = [\mathbf{a}_\sigma, \mathbf{u}_\sigma]$.
8:      Compute guided residual $\mathbf{r} \leftarrow \mathcal{R}(\mathbf{M}_a \odot \mathbf{a}_{\text{obs}} + (1 - \mathbf{M}_a) \odot \mathbf{a}_\sigma, \mathbf{M}_u \odot \mathbf{u}_{\text{obs}} + (1 - \mathbf{M}_u) \odot \mathbf{u}_\sigma)$.
9:      $\hat{\mathbf{x}}_0 \leftarrow D_\theta(\mathbf{x}_\sigma, \sigma, \mathbf{a}_{\text{obs}}, \mathbf{u}_{\text{obs}}, \mathbf{M}_a, \mathbf{M}_u, \mathbf{r})$.          {Compute denoised prediction}
10:     $L \leftarrow \lambda(\sigma) \|\hat{\mathbf{x}}_0 - \mathbf{x}_0\|_H^2$.                           {Compute training loss}
11:     Update parameters $\theta$ by minimizing $L$.
12: **until** converged
13: **return** $D_\theta$

---

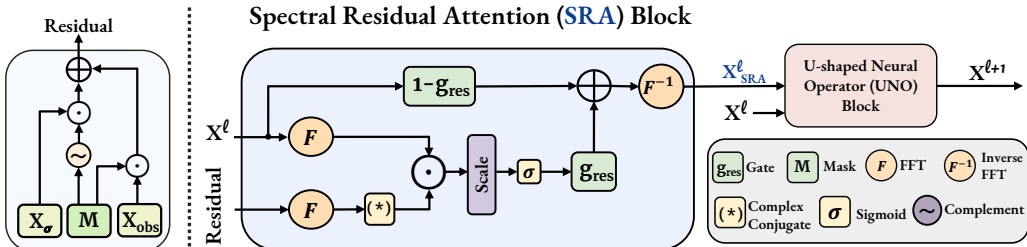

Figure 2: Overview of the PRISMA model architecture. **(Left)** Computation of the ***observation-informed PDE residual***: The model's current estimate ($\mathbf{x}_\sigma$) is mixed with the known observations ($\mathbf{x}_{\text{obs}}$) using the mask ($\mathbf{M}_\mathbf{x}$) to produce a residual field. **(Right)** ***Spectral Residual Attention (SRA) block***: The SRA block operates in the Fourier domain to compute a physics-informed attention mask between the network state ($\mathbf{x}_t^l$) and the residual. This mask is applied in a gated skip-connection, controlled by a learned guidance strength ($g_{\text{res}}$), to produce a modulated state ($\mathbf{x}_{\text{SRA}}^t$) that is then passed to the UNO block.

## 3.4 SPECTRAL RESIDUAL ATTENTION (SRA) BLOCK

Figure 2 shows the architecture of the SRA block that is the core mechanism for injecting physics-based architectural guidance in PRISMA. The goal of SRA is to perform a dynamically adaptive *cross-attention* between the current feature map and the PDE residual, all within the spectral domain. By performing this physics-informed cross-attention, we aim to guide the denoising process toward physically consistent solutions by modulating the frequency spectrum without any loss guidance.

Let $\widetilde{\mathbf{x}}^l = \mathcal{F}(\mathbf{x}^l)$ and $\widetilde{\mathbf{r}} = \mathcal{F}(\mathbf{r})$ be the 2D Fourier transforms of the layer's input feature maps and the corresponding PDE residuals, respectively. The SRA block performs a spectral cross-attention where the PDE residual $\widetilde{\mathbf{r}}^l$ is the **key** and the feature map $\widetilde{\mathbf{x}}^l$ is the **query**. First, a compatibility score $S^l$ measures the phase alignment between them at each 2D frequency mode $\mathbf{k} = (k_1, k_2)$, via their complex inner product. This is calculated as the magnitude of their complex inner product across all channels $C$. This score is then used to create a physics-informed *attention mask* $A^l$ via learnable spectral gain weights $\mathbf{w}_{\text{gain}}^l$, and passing it through a sigmoid activation as follows:

$$S^l(\mathbf{k}) = \frac{1}{\sqrt{C}} \left| \sum_{c=1}^{C} \widetilde{\mathbf{x}}_c^l(\mathbf{k}) \overline{\widetilde{\mathbf{r}}_c^l(\mathbf{k})} \right|, \quad A^l(\mathbf{k}) = \sigma\left(\mathbf{w}_{\text{gain}}^l(\mathbf{k}) \odot S^l(\mathbf{k})\right). \quad (6)$$

This allows the model to learn what frequency modes of the feature map are most informative for error correction based on guidance from PDE residuals. To dynamically control this process based on the relevance of PDE residuals in noisy settings, a scalar *guidance strength* $g_{\text{res}}^l \in [0, 1]$ is learned by an MLP from the spatially-averaged residual $\mathbf{r}_{\text{avg}}^l$ and the diffusion noise embedding $c_\sigma$. Finally, $g_{\text{res}}$ is used to modulate $\widetilde{\mathbf{x}}$ via a skip-connection with the attention mask to produce SRA's output:

$$g_{\text{res}} = \sigma\left(\text{MLP}^l\left([\mathbf{r}_{\text{avg}}^l, c_\sigma]\right)\right), \quad \text{SRA}(\mathbf{x}^l, \mathbf{r}^l) = \mathcal{F}^{-1}\left(\left((1 - g_{\text{res}}^l) + g_{\text{res}}^l A^l\right) \odot \widetilde{\mathbf{x}}^l\right) \quad (7)$$

When the noise level is high, $\mathbf{r}_{avg}^l$ is typically unreliable and the scalar gate $g_{\text{res}}^l$ down-weights PDE residual correction. As the denoising trajectory progresses and the solution becomes cleaner, $g_{\text{res}}^l$ increases, allowing stronger PDE-driven modulation through $A^l(k)$. During training, this entire process of forward propagation is incorporated into the EDM loss computation, teaching the denoiser to produce physically-consistent states. At inference, the model engages in a closed-loop, self-correcting process: it predicts a state, computes the residual based on that state, and then uses that residual to refine its own prediction in the next step of the reverse diffusion trajectory. The complete training loop is detailed in Algorithm 1 while the inference algorithm is in Appendix A.

Table 3: Comparison of different models on three PDE problems with 100% pixels having Unit Gaussian noise corruption, simulating real-world measurement noise (in $L_2$ relative error for all and error rate for Darcy inverse).

| | Steps $(N)$ | Inference Time (s) | Darcy Flow | | Poisson | | Helmholtz | | Avg. Rank $(\downarrow)$ |
|---|---|---|---|---|---|---|---|---|---|
| | | | Forward | Inverse | Forward | Inverse | Forward | Inverse | |
| **FNO** | – | – | 5.3e4% | 1.8e4% | 25.05% | 3.0e7% | 264.8% | 6.1e5% | 5.83 |
| **PINO** | – | – | 1.6e4% | 3.7e4% | 26.64% | 2.3e6% | 449.1% | 1.7e6% | 6.17 |
| **DiffusionPDE** | 2000 | 213.0 | 49.18% | 70.08% | 44.44% | 130.01% | 30.97% | 119.52% | 3.83 |
| **FunDPS** | 200 | 4.72 | 26.9% | 49.39% | 78.36% | 1491.36% | 54.99% | 629.43% | 4.17 |
| **FunDPS** | 500 | 11.8 | 55.09% | 49.62% | 120.0% | 1772.23% | 40.31% | 695.49% | 5.00 |
| **PRISMA (ours)** | 20 | **0.18** | 12.29% | 23.15% | 18.58% | 41.75% | 17.93% | 68.87% | 2.00 |
| **PRISMA (ours)** | 50 | 0.8 | **12.28%** | **23.14%** | **18.35%** | **41.14%** | **16.84%** | **67.93%** | **1.00** |

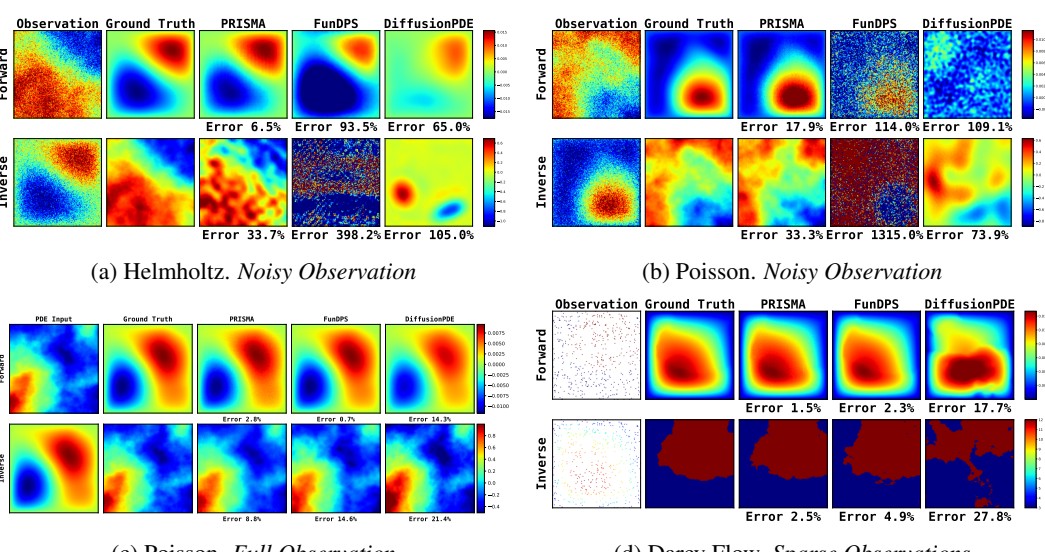

(a) Helmholtz. *Noisy Observation*

(b) Poisson. *Noisy Observation*

(c) Poisson. *Full Observation*

(d) Darcy Flow. *Sparse Observations*

Figure 3: Qualitative results for PRISMA and baseline models on four PDE benchmarks. We evaluate performance under three distinct conditions: (a, b) noisy observations (corrupted by $\mathcal{N}(0, 1)$ Gaussian noise), (c) full, clean observations, and (d) sparse observations (3% of data known). The relative $\ell_2$ error is reported below each prediction (pixel-wise error rate for the Darcy inverse case).

## 4 RESULTS

We demonstrate the efficiency, robustness, and accuracy of PRISMA through a series of quantitative and qualitative experiments. We evaluate PRISMA on three distinct PDE problems, focusing on its performance in noisy, fully-observed, and sparsely-observed settings. Additional experiments and implementation details are provided in the Appendix (see sections B, D, E and I).

### 4.1 EXPERIMENTAL SETUP

**Dataset & Tasks:** We validate our approach on three PDE problems from the dataset of Huang et al. (2024), which is characterized by multi-scale features, complex boundary conditions, and nonlinear dynamics. We evaluate both forward and inverse problems in three regimes: (1) *Full Observation*, with complete and clean data; (2) *Sparse Observation*, where only 3% of the input field is known (e.g., initial condition or solution); and (3) a challenging *Noisy* setting, where the observed data is corrupted by additive Gaussian noise sampled from $\mathcal{N}(0, 1)$. For dataset details, see Appendix C.

**Training & Inference:** Our implementation builds on the EDM-FS framework (Beckham, 2024) using a U-shaped neural operator (Rahman et al., 2022) as the denoiser $D_\theta$ and adapting the pro-

Table 4: Ablation of residual strategies across Helmholtz (full, noisy, sparse) for both forward/inverse problems ($L_2$ relative error, shown as percentages). All results are on 64×64 resolution with 20 steps.

| | Full Helmholtz | | Noisy Helmholtz | | Sparse Helmholtz | |
|---|---|---|---|---|---|---|
| | **Forward** | **Inverse** | **Forward** | **Inverse** | **Forward** | **Inverse** |
| **PRISMA** *(w/o PDE res)* | $\mathbf{3.15_{0.05}}\%$ | $15.67_{0.34}\%$ | $\mathbf{29.0_{0.20}}\%$ | $111.87_{2.12}\%$ | $36.2_{0.15}\%$ | $66.75_{0.19}\%$ |
| **PRISMA** *(PDE res with concat)* | $28.7_{0.9}\%$ | $51.55_{5.75}\%$ | $34.5_{1.9}\%$ | $97.09_{1.32}\%$ | $94.2_{0.5}\%$ | $124.7_{0.24}\%$ |
| **PRISMA** *(PDE res with SRA)* **(ours)** | $3.34_{0.14}\%$ | $\mathbf{12.47_{0.11}}\%$ | $30.35_{0.45}\%$ | $\mathbf{91.85_{1.35}}\%$ | $\mathbf{30.33_{0.12}}\%$ | $\mathbf{60.96_{0.09}}\%$ |

cess for functional noise. Our model's parameter count is comparable to key baselines, with full implementation details provided in Appendix B.

**Baselines & Evaluation Metrics:** We compare against deterministic solvers including FNO (Li et al., 2020), PINO (Li et al., 2024), and PINN (Raissi et al., 2019a), as well as state-of-the-art diffusion methods like DiffusionPDE (Huang et al., 2024) and FunDPS (Yao et al., 2025). Performance across all tasks is measured using the relative $L_2$ error. For the Darcy flow inverse problem, we report the classification error rate.

## 4.2 RESULTS ON NOISY OBSERVATIONS

We first evaluate PRISMA's robustness in the challenging noisy observation setting, mimicking real-world measurement noise. The results, summarized in Table 3, demonstrate that PRISMA achieves state-of-the-art performance across most of the equations. For instance, in the Darcy Flow forward problem, PRISMA with 50 steps achieves an $L_2$ relative error of 12.28%, outperforming the baselines (55.09% and 49.18%). We see a similar trend in other equations as well. While our primary focus is the challenging noisy setting, PRISMA also demonstrates competitive performance under full and sparse observation settings (Tables 6 and 7), while requiring far fewer sampling steps. For a qualitative comparison, we provide key visualizations in Figure 3, with more extensive results available in Appendix I.

## 4.3 INFERENCE EFFICIENCY

A key advantage of PRISMA is its significant inference efficiency while having competitive accuracy, compared to the current state-of-the-art. As shown in Table 3 and visualized in Figure 4, PRISMA occupies the optimal low-error, low-time quadrant. We attribute this inference speedup as a direct result of our unified conditioning framework and the strong physical guidance from the SRA block. This enables a gradient-descent free sampling process during inference, whereas DiffusionPDE and FunDPS rely on computationally expensive PDE-based inference sampling. By directly conditioning on the input, PRISMA converges in just **20-50 steps**, while DiffusionPDE and FunDPS require 200 to 2000 steps.

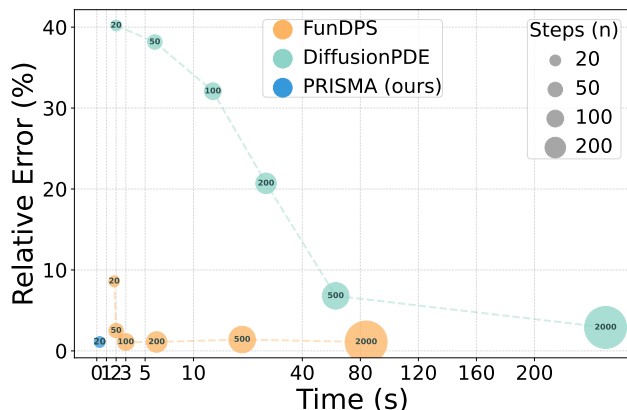

Figure 4: Inference time vs. accuracy trade-off on the Darcy Flow forward problem under full observations. Point size corresponds to the number of denoising steps.

When they are also restricted to 20 steps, their performance degrades significantly as shown in Tables 10, 11. PRISMA has an overall inference speed-up of **15x to 250x** (seconds per sample) compared to competing diffusion-based PDE solvers.

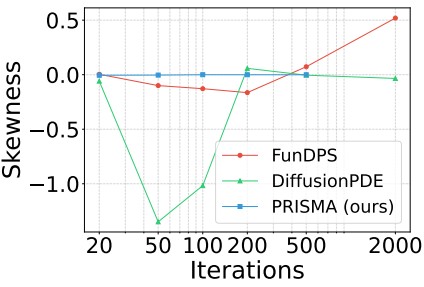 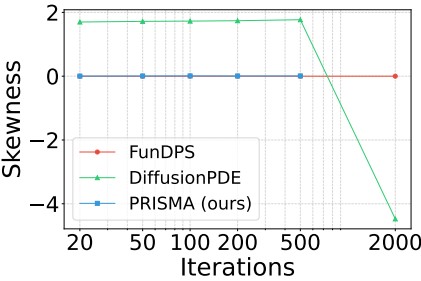

Figure 5: Skewness of the PDE residual field plotted against inference iterations for the Poisson equation. (*Left*: Forward problem, *Right*: Inverse problem)

### 4.4 ANALYSIS OF MODEL ROBUSTNESS AND RESIDUALS

To further probe our model's capabilities, we analyzed its performance under varying noise levels and examined the statistical properties of its PDE residuals.

**Impact of PDE Residual Guidance:** A key design choice in PRISMA is the architectural integration of PDE residuals via the SRA block. To justify this choice, we compare three variants: (1) **No PDE Residual**, a baseline without physics guidance; (2) **Concatenation**, a simple channel-wise PDE residual concatenation; and (3) **PDE Residual with SRA**, where the PDE residual is input to the SRA block. Table 4 highlights that the PDE residual with SRA block outperforms both simple channel-wise concatenation of the PDE residual and an unguided baseline.

**Robustness to Varying Noise Levels** We tested PRISMA on the Helmholtz equation with varying intensities of Gaussian noise ($\sigma$). As plotted in Figure 6, PRISMA consistently maintains a lower error rate than FunDPS and DiffusionPDE across all noise levels for both forward and inverse tasks. Notably, while the error for baseline models increases sharply with noise, PRISMA's performance degrades more gracefully, highlighting its enhanced stability in high-noise regimes.

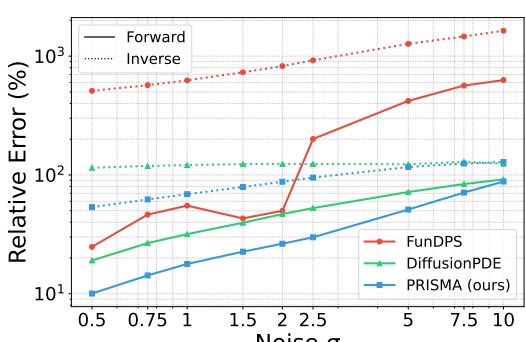

Figure 6: Relative error vs. input noise level ($\sigma$) for the Helmholtz forward (solid lines) and inverse (dotted lines) problems.

**Fidelity to Physical Constraints** A well-trained model should produce PDE residuals with skewness and kurtosis values close to zero. Figure 5 shows the skewness residual statistics for the Poisson problem. PRISMA's residuals (blue) exhibit near-zero skewness across different inference sampling iterations. We observe a similar trend for kurtosis (see Appendix Figure 7).

## 5 CONCLUSION & FUTURE WORK

In this work, we introduced PRISMA, a novel conditional diffusion neural operator that fundamentally challenges the prevailing paradigm of using PDE residuals as external loss terms for guidance. Our approach centers on architectural guidance, embedding physical constraints directly into the model as learnable features via our novel Spectral Residual Attention (SRA) block. We condition the model with task & sparsity masks during training to perform both unconditional and conditional generation. This design enables two critical advantages: first, it facilitates an entirely gradient-descent free sampling process during inference, eliminating slow and unstable test-time optimization. Second, it allows the creation of a single, unified model capable of seamlessly solving both forward and inverse problems across the full spectrum of full, sparse, and noisy observation regimes. Our experiments validate the effectiveness of our approach, demonstrating inference speedups of 15x to 250x (seconds per sample) relative to state-of-the-art diffusion-based methods while delivering competitive or superior accuracy and robustness especially against noisy observations.

Our work opens several promising avenues for future research. A primary direction is the extension of PRISMA to spatio-temporal problems, which would involve adapting the architecture to handle time-dependent dynamics. Another key step is to generalize the model to operate on complex geometries and irregular meshes, moving beyond grid-based problems.

## REPRODUCIBILITY STATEMENT

To ensure reproducibility, we provide the complete source code in the supplementary materials. We used publicly available datasets described in Appendix C, along with all hyperparameters and experimental details are described in Appendix B.

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

APPENDICES

## A  INFERENCE SAMPLING ALGORITHM

Algorithm 2 details the inference sampling procedure for PRISMA, which employs a 2nd-order solver to iteratively generate a physically consistent solution from a random field. A key innovation of this process is the use of an observation-guided PDE residual, which is re-computed and fed into the denoising model at every step of the sampling process.

The core of the algorithm lies in the guided self-correction step(Lines 4 and 9). At each iteration $i$, the PDE residual $r$ is not computed on the model's raw prediction alone. Instead, it is calculated on a composite field $(1 - M) \odot x_i + M \odot x_{\text{obs}}$, where $M$ is a binary mask. This formulation ensures that for the known parts of the domain (where $M = 1$), the residual calculation is grounded by the true observations $x_{\text{obs}}$. For the unknown parts (where $M = 0$), it uses the model's current estimate $x_i$.

For instance, consider a forward problem with full observation, where the state $x = [\mathbf{a}, \mathbf{u}]$ consists of the known input coefficients $\mathbf{a}$ and the unknown solution $\mathbf{u}$. In this case, the mask $M = [M_{\mathbf{a}}, M_{\mathbf{u}}]$ would have $M_{\mathbf{a}}$ as a matrix of ones (fully observed) and $M_{\mathbf{u}}$ as a matrix of zeros (fully unobserved). Consequently, the composite field used to calculate the residual becomes $[\mathbf{a}_{\text{obs}}, \mathbf{u}_i]$. This means the PDE residual operator $\mathcal{R}$ evaluates physical consistency based on the true input $\mathbf{a}$ and the model's current prediction for the solution $u_i$.

This guided residual $r$ acts as an explicit, spatially-varying map of physical inconsistency, which is then passed as a direct input to the denoising operator $D_\theta$ (Lines 5 and 10). By providing this physical guidance at every step of the predictor-corrector solver, the model is continuously steered toward solutions that are not only consistent with the initial observations but also compliant with the governing PDE, enabling fast, gradient-descent free convergence.

---

**Algorithm 2** PRISMA Inference with 2nd-Order Solver and Guided Residual

---

**Require:** Observations $\mathbf{x}_{\text{obs}} = [\mathbf{a}_{\text{obs}}, \mathbf{u}_{\text{obs}}]$, masks $\mathbf{M} = [\mathbf{M}_a, \mathbf{M}_u]$, PDE residual operator $\mathcal{R}$, denoising diffusion operator $D_\theta$, variance schedule $\{\sigma_i\}_{i=0}^N$ with $\sigma_0 = 0$.

1: $\mathbf{x}_N \sim \mathcal{N}(0, \mathbf{C})$                  {Initialize from GRF}
2: **for** $i = N, \ldots, 1$ **do**
3:   Let current state be $\mathbf{x}_i = [\mathbf{a}_i, \mathbf{u}_i]$.           {*Predictor Step*}
4:   Compute guided residual $\mathbf{r} \leftarrow \mathcal{R}((1 - \mathbf{M}) \odot \mathbf{x}_i + \mathbf{M} \odot \mathbf{x}_{\text{obs}})$.   {Guided self-correction}
5:   Predict clean state $\hat{\mathbf{x}}_0 \leftarrow D_\theta(\mathbf{x}_i, \sigma_i, \mathbf{x}_{\text{obs}}, \mathbf{M}, \mathbf{r})$.
6:   $\mathbf{d}_i \leftarrow (\mathbf{x}_i - \hat{\mathbf{x}}_0)/\sigma_i$.           {Evaluate derivative $d\mathbf{x}/d\sigma$}
7:   $\mathbf{x}'_{i-1} \leftarrow \mathbf{x}_i + (\sigma_{i-1} - \sigma_i)\mathbf{d}_i$.         {Take an Euler step}
8:   **if** $\sigma_{i-1} \neq 0$ **then**            {*Corrector Step*}
9:    Compute new guided residual $\mathbf{r}' \leftarrow \mathcal{R}((1 - \mathbf{M}) \odot \mathbf{x}'_{i-1} + \mathbf{M} \odot \mathbf{x}_{\text{obs}})$.   {Guided self-correction}
10:    Predict clean state $\hat{\mathbf{x}}'_0 \leftarrow D_\theta(\mathbf{x}'_{i-1}, \sigma_{i-1}, \mathbf{x}_{\text{obs}}, \mathbf{M}, \mathbf{r}')$.
11:    $\mathbf{d}'_i \leftarrow (\mathbf{x}'_{i-1} - \hat{\mathbf{x}}'_0)/\sigma_{i-1}$.
12:    $\mathbf{x}_{i-1} \leftarrow \mathbf{x}_i + (\sigma_{i-1} - \sigma_i) \cdot (\frac{1}{2}\mathbf{d}_i + \frac{1}{2}\mathbf{d}'_i)$.    {Apply 2nd-order correction}
13:   **else**
14:    $\mathbf{x}_{i-1} \leftarrow \mathbf{x}'_{i-1}$.
15:   **end if**
16: **end for**
17: **return** $\mathbf{x}_0$

---

# B IMPLEMENTATION DETAILS

We adopt a 4-level U-shaped neural operator architecture (Rahman et al., 2022) as the denoiser $D_\theta$, which has 64M parameters, similar to DiffusionPDE's and FunDPS's network size. The network is trained using 50,000 training samples for 200 epochs on 2 NVIDIA A100 GPUs with a Batch size of 90 per GPU. The code-base is built upon Beckham (2024)'s implementation of Denoising Diffusion Operators (DDO). The hyperparameters we used for training and inference are listed in Table 5 and were taken from Beckham (2024)'s DDO implementation. We source the quantitative results of deterministic baselines for the full and noisy cases from DiffusionPDE's (Huang et al., 2024) table. We faced reproducibility issues for DiffusionPDE, also observed by Yao et al. (2025). We had correspondence with DiffusionPDE's authors about this and have rerun all their experiments for the Full and Sparse cases, and the tables 6 and 7 show our reproduced results for their method.

**Residual-aware Guidance Strength MLP:** Each SRA block uses a small two-layer MLP to predict the scalar guidance weight $g_{\text{res}} \in [0, 1]$. The inputs to this MLP are: (i) the diffusion/timestep embedding $c_\sigma$ (of length $E$, e.g., $E{=}256$), and (ii) the spatial mean of the residual $r_{\text{avg}}$ (a scalar). We concatenate these to form a tensor of shape $B \times (E{+}1)$, where $B$ is the batch size, and pass it through

$$\text{Linear}(E{+}1 \rightarrow E) \ \rightarrow \ \text{ReLU} \ \rightarrow \ \text{Linear}(E \rightarrow 1) \ \rightarrow \ \text{Sigmoid},$$

yielding a $B \times 1$ output used as the skip-connected gating weight within the SRA block. Each SRA block contains its own distinct MLP.

**Speed comparison** All the experiments are conducted using a single NVIDIA A100 GPU. To determine per-sample inference time, we averaged batch inference time over 10 runs and divided by the batch size.

# C DATASETS

We evaluate our approach on three benchmark PDE problems: including Darcy Flow, Poisson and Helmholtz. Each PDE is considered in both forward ($a \rightarrow u$) and Inverse ($u \rightarrow a$) settings. We use the same data as used in DiffusionPDE (Huang et al., 2024) which consists of 50,000 training and 1000 test samples for each PDE, at both $64{\times}64$ and $128{\times}128$ resolution. All models are trained and evaluated at $128{\times}128$, and some ablations when stated are on the $64{\times}64$ resolution. For quantitative evaluation, we report the mean relative $L^2$ error between the predicted and true solutions, except for the inverse Darcy Flow problem, where we use the binary error rate.

Table 5: Hyperparameters

| Hyperparameter | Value |
|---|---|
| learning_rate | 0.0001 |
| learning_rate_warmup | 50 epochs |
| ema_half_life | 5 epochs |
| dropout | 0.13 |
| rbf_scale | 0.05 |
| sigma_max | 80 |
| sigma_min | 0.002 |
| rho | 7 |

**Darcy Flow Equation**   Darcy flow describes the movement of fluid through a porous medium. It is governed by the equation:

$$-\nabla \cdot \big(a(x)\nabla u(x)\big) = g(x), \quad x \in \Omega, \tag{8}$$

where the domain is $\Omega = (0,1)^2$, with $g(x) = 1$ (constant forcing) and zero Dirichlet boundary conditions. The coefficient function is sampled as $a \sim h_\#\mathcal{N}\big(0, (-\Delta + 9\mathbf{I})^{-2}\big)$, following (Huang et al., 2024; Yao et al., 2025). The mapping $h$ is defined piecewise, taking the value 12 for positive inputs and 3 otherwise.

**Poisson Equation.**   We consider the Poisson equation on the unit square $\Omega = (0,1)^2$ which describes steady-state diffusion processes:

$$\nabla^2 \mathbf{u}(x) = \mathbf{a}(x), \quad x \in \Omega, \tag{9}$$

subject to homogeneous Dirichlet boundary conditions:

$$\mathbf{u}(x) = 0, \quad x \in \partial\Omega. \tag{10}$$

The source term $\mathbf{a}(x)$ is a binary field, obtained by thresholding a Gaussian random field at zero:

$$\mathbf{a}(x) = \mathbf{1}_{Z(x)>0}, \quad Z \sim \mathcal{N}\big(0, (-\Delta + 9\mathbf{I})^{-2}\big), \tag{11}$$

where $\mathbf{1}.$ is the indicator function. The PDE residual, which serves as a physical constraint for the model, is defined as,

$$f(x) = \nabla^2 \mathbf{u}(x) - \mathbf{a}(x). \tag{12}$$

**Helmholtz Equation.**   We consider the static inhomogeneous Helmholtz equation on the domain $\Omega = (0,1)^2$:

$$\Delta u(x) + k^2 u(x) = a(x), \quad x \in \Omega, \tag{13}$$

subject to homogeneous Dirichlet boundary conditions:

$$u(x) = 0, \quad x \in \partial\Omega, \tag{14}$$

where $k = 1$. This equation describes wave propagation in heterogeneous media. The coefficient function $a(x)$ is generated as a piecewise-constant field sampled from a Gaussian random field. Unlike the Poisson case, where the source term is thresholded to a binary field, the Helmholtz coefficients can take a range of values across different regions, giving heterogeneous variation. The solution $u(x)$ is computed using second-order finite differences, with zero boundary enforced via the same mollifier as in (Huang et al., 2024).

## D   RESULTS

### D.1   FULL OBSERVATION

Table 6 compares different models across three PDE problems for the full observation setting. Our method achieves competitive performance, achieving an average rank of 2.1 across all tasks, followed by FunDPS and then PINO. In addition to accuracy, our approach demonstrates significant efficiency: compared to other diffusion-based models, it is 15x to 250x (seconds per sample) faster during inference. Note that the baseline values for DiffusionPDE are taken from the original paper. Reproduced results for DiffusionPDE are reported here due to reproducibility issues, as also noted in FunDPS.

Table 6: Comparing different models on three PDE problems (in $L_2$ relative error) under Full Observation. * denotes results reproduced using authors' released code and checkpoints. **Best** is boldened, second-best underlined.

| | Steps $(N)$ | Inference Time | Darcy Flow | | Poisson | | Helmholtz | | Avg. Rank |
| --- | --- | --- | --- | --- | --- | --- | --- | --- | --- |
| | | | Forward | Inverse | Forward | Inverse | Forward | Inverse | |
| **PINO** | – | – | 4% | **2.1%** | 3.7% | **10.2%** | 4.9% | **4.9%** | **2.50** |
| **DeepONet** | – | – | 12.3% | 8.4% | 14.3% | 29% | 17.8% | 28.1% | 7.83 |
| **PINNs** | – | – | 15.4% | 10.1% | 16.1% | 28.5% | 18.1% | 29.2% | 8.67 |
| **FNO** | – | – | 5.3% | 5.6% | 8.2% | 13.6% | 11.1% | 5.0% | 5.33 |
| **DiffusionPDE*** | 2000 | 213 | 2.9% | 13% | 15.27% | 21.21% | 10.9% | 18.97% | 6.83 |
| **FunDPS*** | 200 | 4.72 | 1.1% | 4.2% | **0.7%** | 23.32% | **1.08%** | 18.48% | 3.83 |
| **FunDPS*** | 500 | 11.78 | 1.4% | 3.0% | 0.84% | 19.84% | **1.08%** | 13.88% | 3.25 |
| **FunDPS** | 2000 | 50.4 | **0.9%** | **2.1%** | – | – | – | – | — |
| **PRISMA (ours)** | 20 | **0.18** | 1.1% | 3.8% | 4.58% | 10.9% | 8.12% | 11.03% | 3.92 |
| **PRISMA (ours)** | 50 | 0.8 | 1.05% | 3.79% | 4.0% | 10.7% | 7.11% | 10.76% | 2.83 |

## D.2 SPARSE OBSERVATIONS

Table 7 presents a comparison of PRISMA against deterministic and generative baselines under the challenging sparse observation setting. While FunDPS (at 500 steps) achieves the highest accuracy, PRISMA demonstrates a superior balance of performance and efficiency. Our model delivers competitive accuracy, comparable to the top-performing generative methods, but at a fraction of the computational cost. Notably, PRISMA at 20 steps is over 65 times faster than the most accurate FunDPS configuration and over 1000 times faster than DiffusionPDE. This highlights PRISMA's significant advantage in inference speed, making it a practical and efficient choice for applications where rapid predictions are critical.

Table 7: Comparison of models on **Darcy Flow** (in $L_2$ relative error) under Sparse Observation (97% pixels). **Bold** denotes best, underline second-best.

| | Steps $(N)$ | Darcy Flow | | Average Rank |
| --- | --- | --- | --- | --- |
| | | Forward | Inverse | |
| **FNO** | – | 28.2% | 49.3% | 7.00 |
| **PINO** | – | 35.2% | 49.2% | 7.00 |
| **DeepONet** | – | 38.3% | 41.1% | 7.00 |
| **PINN** | – | 48.8% | 59.7% | 9.00 |
| **DiffusionPDE** | 2000 | 6.07% | 7.87% | 5.00 |
| **FunDPS** | 200 | 2.88% | 6.78% | 2.00 |
| **FunDPS** | 500 | **2.49%** | **5.18%** | **1.00** |
| **PRISMA (ours)** | 20 | 3.13% | 6.80% | 3.75 |
| **PRISMA (ours)** | 50 | 3.11% | 6.80% | 3.25 |

# E ABLATIONS

## E.1 ABLATION ON THE NUMBER OF INFERENCE STEPS

To analyze the convergence behavior of PRISMA, we evaluate its performance across a range of inference steps $(N)$ from 20 to 500. This sensitivity analysis is conducted for both the challenging noisy observation setting (Table 8) and the ideal full observation setting (Table 9). The results clearly demonstrate that PRISMA converges remarkably quickly. In both scenarios, performance saturates early, with minimal to no improvement observed beyond 20-50 steps. This rapid convergence justifies our use of a small number of steps for inference, as it provides an optimal balance between computational efficiency and accuracy.

Table 8: Performance of PRISMA (relative $L_2$ error %) on noisy observation tasks as the number of inference steps ($N$) is varied.

| Model | Steps ($N$) | Darcy Flow | | Poisson | | Helmholtz | |
|---|---|---|---|---|---|---|---|
| | | Fwd | Inv | Fwd | Inv | Fwd | Inv |
| **PRISMA** | 20 | 12.10 | 22.93 | 18.20 | 41.46 | 17.65 | 68.43 |
| **PRISMA** | 50 | 12.06 | 22.97 | 18.10 | 40.90 | 16.50 | 67.50 |
| **PRISMA** | 100 | 12.06 | 22.88 | 18.00 | 40.90 | 16.50 | 67.50 |
| **PRISMA** | 200 | 12.09 | 22.91 | 17.70 | 40.90 | 16.50 | 67.50 |
| **PRISMA** | 500 | 12.06 | 23.03 | 18.00 | 40.80 | 16.40 | 67.50 |

Table 9: Performance of PRISMA (relative $L_2$ error %) on full observation tasks as the number of inference steps ($N$) is varied.

| Model | Steps ($N$) | Darcy Flow | | Poisson | | Helmholtz | |
|---|---|---|---|---|---|---|---|
| | | Fwd | Inv | Fwd | Inv | Fwd | Inv |
| **PRISMA** | 20 | 1.10 | 3.80 | 4.58 | 10.90 | 8.12 | 11.03 |
| **PRISMA** | 50 | 1.05 | 3.79 | 4.00 | 10.70 | 7.11 | 10.76 |
| **PRISMA** | 100 | 1.04 | 3.78 | 4.00 | 10.60 | 6.94 | 10.75 |
| **PRISMA** | 200 | 1.03 | 3.78 | 4.00 | 10.60 | 6.92 | 10.70 |
| **PRISMA** | 500 | 1.04 | 3.78 | 4.00 | 10.60 | 6.80 | 10.70 |

### E.2 COMPARISON AT 20 STEPS

To highlight the inference efficiency of our method, we conduct a direct comparison where all models are restricted to just 20 sampling steps, a regime where PRISMA excels. While the optimal performance for FunDPS and DiffusionPDE is achieved at much higher step counts (200–500 and 2000 steps, respectively), this analysis serves as a stress test to evaluate per-step convergence speed. As shown in Table 10, PRISMA consistently achieves low error rates across all tasks in the noisy setting. In contrast, the baseline models produce significantly higher errors, with FunDPS often failing to converge entirely. We see a similar trend for the full observation case in Table 11. This demonstrates that PRISMA's architectural guidance enables a much more rapid convergence to accurate, physically consistent solutions.

Table 10: Comparative performance (relative $L_2$ error %) on noisy observation tasks with 20 inference steps.

| Model | Steps ($N$) | Darcy Flow | | Poisson | | Helmholtz | |
|---|---|---|---|---|---|---|---|
| | | Fwd | Inv | Fwd | Inv | Fwd | Inv |
| **FunDPS** | 20 | 99.99 | 73.00 | 99.99 | 255.09 | 99.99 | 178.09 |
| **DiffusionPDE** | 20 | 37.16 | 70.85 | 125.15 | 148.97 | 123.44 | 133.35 |
| **PRISMA (ours)** | 20 | 12.29 | 23.15 | 18.58 | 41.75 | 17.93 | 68.87 |

Table 11: Comparative performance (relative $L_2$ error %) on full observation tasks with 20 inference steps.

| Model | Steps ($N$) | Darcy Flow | | Poisson | | Helmholtz | |
|---|---|---|---|---|---|---|---|
| | | Fwd | Inv | Fwd | Inv | Fwd | Inv |
| **FunDPS** | 20 | 8.88 | 17.75 | 9.755 | 39.15 | 10.08 | 39.39 |
| **DiffusionPDE** | 20 | 30.99 | 69.82 | 95.26 | 123.21 | 111.77 | 101.01 |
| **PRISMA (ours)** | 20 | 1.10 | 3.80 | 4.58 | 10.90 | 8.12 | 11.03 |

### E.3 FIDELITY TO PHYSICAL CONSTRAINTS

To assess the physical consistency of the generated solutions, we analyze the statistical properties of the PDE residual fields. A well-trained model should produce residuals with a distribution close

Table 12: SRA Ablations (comparing $L_2$ error %) of gating and attention mechanisms on 64×64 resolution

| | **Steps** $(N)$ | **Full Helmholtz** | | **Noisy Helmholtz** | |
| --- | --- | --- | --- | --- | --- |
| | | **Forward** | **Inverse** | **Forward** | **Inverse** |
| **No Gating** | 20 | 3.8 | 28.91 | 33.8 | 97.5 |
| **Multiplicative Gate** *(no skip)* | 20 | 3.3 | 12.98 | **24.7** | 115.07 |
| **Only Mag of PDE Residual** *(w/o attention)* | 20 | 3.29 | 13.42 | 25.44 | 115.26 |
| **Attention** *(w/o Phase)* | 20 | 6.5 | 24.77 | 34 | 102.19 |
| **Without Guided PDE Residual** | 20 | 3.52 | 14.26 | 31.26 | 98.99 |
| **Ours (PDE Res SRA)** | 20 | 3.48 | **12.58** | 30.8 | **93.2** |

to a standard normal. A kurtosis value close to zero indicates that the errors are random and well-behaved, rather than systematic.

Figure 7 plots the kurtosis of the PDE residual for the Poisson problem against the number of inference iterations. The results show that PRISMA's residual kurtosis (blue line) is consistently stable and near zero across all iterations for both the forward and inverse problems. In contrast, DiffusionPDE (green line) exhibits highly unstable and large kurtosis values, indicating that its solutions contain significant non-Gaussian errors or physical inconsistencies. This analysis demonstrates that PRISMA's architectural guidance leads to more physically robust solutions.

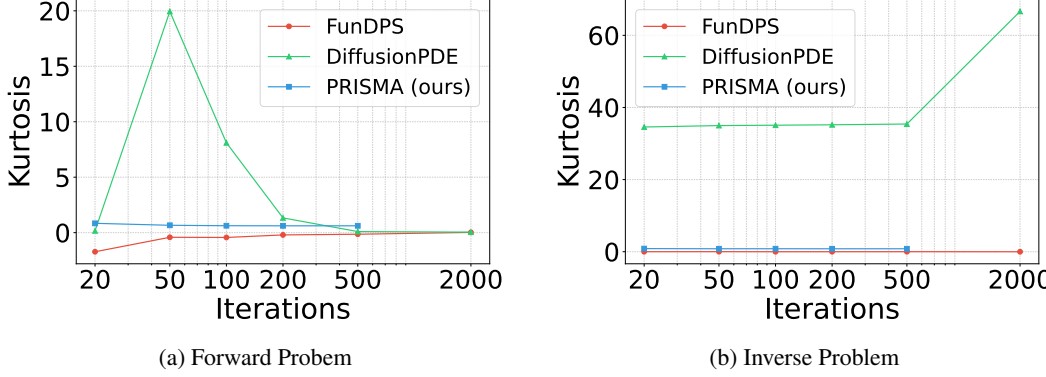

(a) Forward Probem    (b) Inverse Problem

Figure 7: Kurtosis of the PDE residual field plotted against the number of inference iterations for the Poisson problem.

### E.4    SRA ABLATIONS

**SRA Ablations:** We further analyze the Spectral Residual Attention (SRA) block by running ablations at 64×64 with $N$=20 inference steps Table 12. We compare the following setups: (1) **No Gating** where $g_{res}$=1 effectively; (2) **Multiplicative gate** where there is no weighted skip connection, (3) **Only magnitude** where we replace cross-attention with a per-mode weight from $|\hat{r}(k)|$, (4) **Attention w/o phase** which considers the magnitude only and is phase-blind ($|\hat{x}(k)| \cdot |\hat{r}(k)|$), (5) **Without guided PDE residual** where we do not guide the PDE residual as described in Appendix Section A, and **Ours (PDE Res SRA)**. We observe that the choice of gating matters, the skip-connected scalar gate stabilizes residual injection and helps notably in noisy inverse settings. Furthermore, we see that phase-aware spectral attention outperforms magnitude-only/phase-blind variants, indicating that per-mode phase alignment provides useful physics signal and lastly residual guidance helps SRA consistently improve over no-guidance on inverse tasks while remaining competitive on forward tasks, more so in noisy observation settings.

## F  SPATIOTEMPORAL FIELD RECONSTRUCTION

We evaluate PRISMA on a time-dependent PDE to demonstrate its capability to recover the full solution trajectory over the interval $[0, T]$ given spatially sparse but temporally continuous sensor observations. We focus on the 1D dynamic Burgers' equation, where PRISMA learns the joint spatiotemporal distribution $u_{0:T}$ using a 2D diffusion model. In our comparison, PRISMA outperforms DiffusionPDE, achieving a test relative error of 9.33% versus 10.39%. Figure 8 show qualitative comparison of the solutions generated by PRISMA and DiffusionPDE on Burgers' equation.

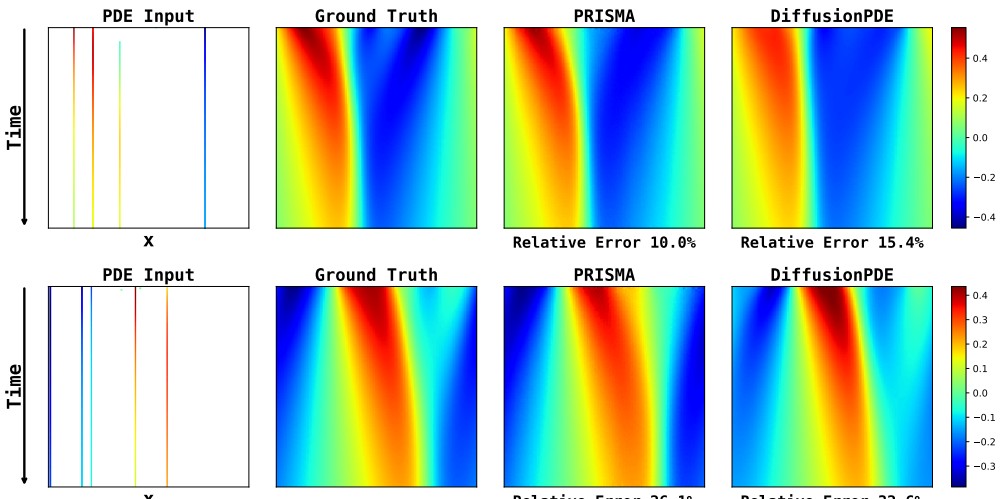

Figure 8: Prediction (*for two samples*) Visualization on Burgers' equation

## G  SUPER-RESOLUTION ANALYSIS

Table 13: Performance metrics at different resolutions. Average batch time reported for batch size=50

| Resolution | Avg. Batch Time (s) | Avg. Per-Sample Time (s) | Avg. Peak GPU (reserved) Mem (GB) |
|---|---|---|---|
| 64x64 | $12.140 \pm 0.945$ | $0.242 \pm 0.018$ | $4.117 \pm 0.09$ |
| 128x128 | $20.036 \pm 0.326$ | $0.400 \pm 0.006$ | $15.383 \pm 0.804$ |
| 256x256 | $60.892 \pm 0.580$ | $1.217 \pm 0.011$ | $62.602 \pm 0.804$ |
| 512x512 | $242.489 \pm 0.423$ | $4.849 \pm 0.008$ | $107.114 \pm 0.399$ |

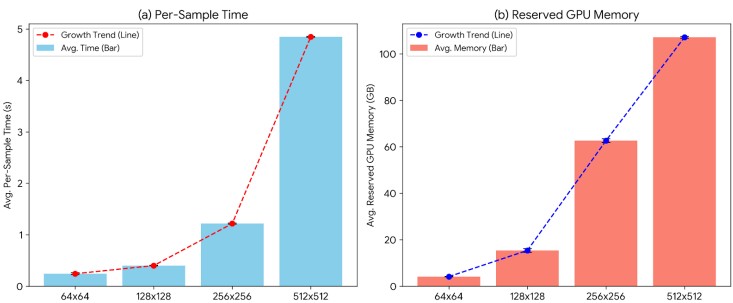

Figure 9: Performance metrics (Per-Sample Time and Peak Reserved GPU Memory) as a function of inference resolution.

We evaluate PRISMA's ability to generalize to higher spatial resolutions at inference using a model trained only on 64×64 resolution. For each target resolution (64, 128, 256, and 512), we upsample

the 64×64 PDE input and run the same 20-step PRISMA model, without retraining the architecture, to generate the corresponding solution. Qualitative forward and inverse results are shown in Figures 10, 11, and 12, and the corresponding runtime and memory measurements are reported in Table 13. Figure 9 visualizes how inference cost grows with resolution, specifically the per-sample time increases smoothly from 0.24s → 4.85s, and reserved GPU memory rises from 4 GB → 107 GB. While the 512×512 case incurs a significantly higher memory footprint, PRISMA remains stable across all resolutions tested.

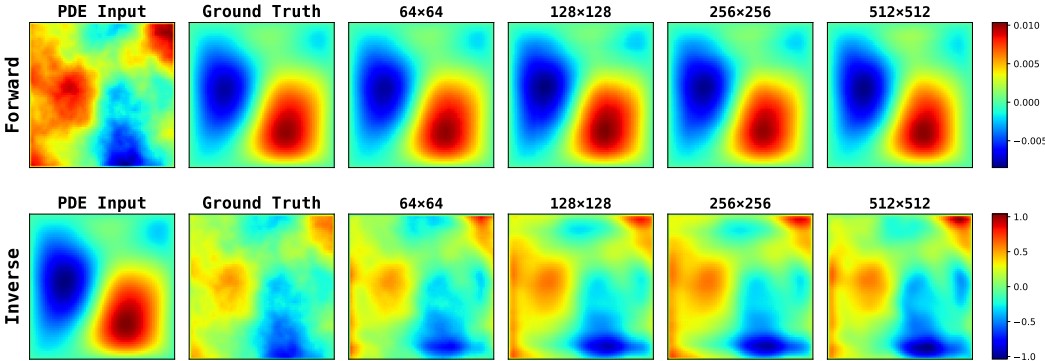

Figure 10: Super-resolution results on Helmholtz

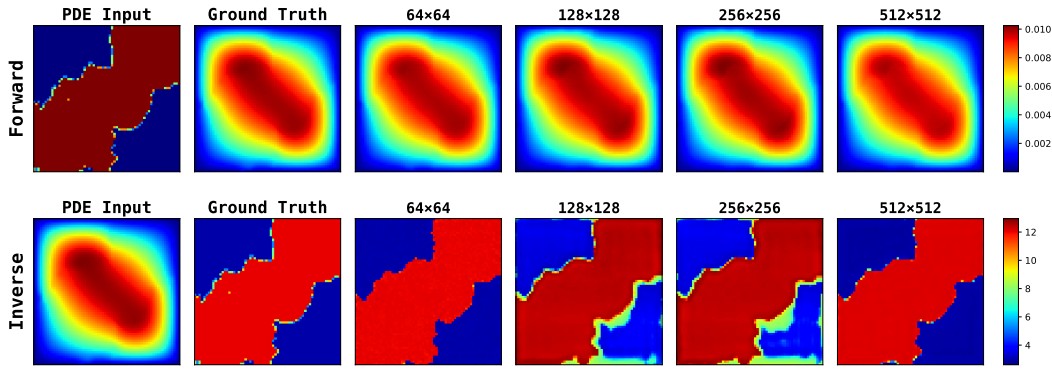

Figure 11: Super-resolution results on Darcy

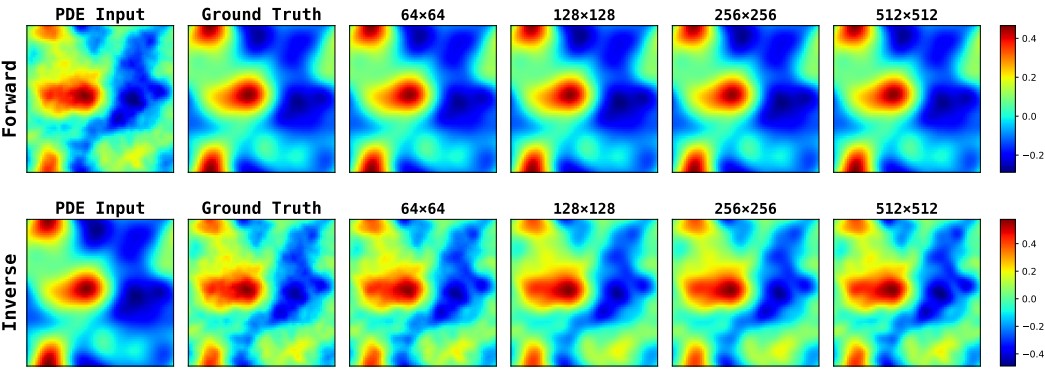

Figure 12: Super-resolution results on Navier-Stokes (NB)

## H  NOISE ROBUSTNESS EVALUATION

We evaluate PRISMA and all baselines under varying levels of additive measurement noise. For each sample, we corrupt a fixed percentage of pixels (90%, 50%, 30%, 10%) with unit-variance Gaussian

noise, keeping the remaining pixels clean. This setup simulates real-world sensor degradation where only part of the field is corrupted. The diffusion models (including PRISMA) are not trained with any noise augmentation beyond the intrinsic diffusion noising, all methods are evaluated out-of-distribution. Tables 14–17 report results across five PDEs in forward and inverse settings. Across all noise levels, classical operator-based models (FNO/PINO) degrade sharply, while diffusion models remain substantially more stable. FunDPS benefits from per-sample DPS optimization but becomes sensitive at high noise levels and in inverse regimes. PRISMA shows consistent performance across all noise ratios, even when a large majority of pixels are corrupted.

Table 14: Comparison of different models on three PDE problems with 90% pixels having Unit Gaussian noise corruption, simulating real-world measurement noise (in $L_2$ relative error for all and error rate for Darcy inverse).

| | Steps $(N)$ | Darcy Flow | | Poisson | | Helmholtz | |
|---|---|---|---|---|---|---|---|
| | | Forward | Inverse | Forward | Inverse | Forward | Inverse |
| FNO | – | 72.84% | 8974.61% | 80.48% | 1.44e07% | 80.04% | 2.92e05% |
| PINO | – | 70.42% | 17500.2% | 80.44% | 1.12e06% | 107.15% | 8.74e05% |
| DiffusionPDE | 2000 | 36.89% | 70.04% | 44.51% | 129.08% | 23.2% | 113.49% |
| FunDPS | 200 | 10.81% | 48.24% | 16.07% | 957.87% | 16.96% | 622.6% |
| PRISMA | 20 | **9.65%** | **22.83%** | **15.69%** | **40.34%** | **16.16%** | **66.93%** |

Table 15: Comparison of different models on three PDE problems with 50% pixels having Unit Gaussian noise corruption, simulating real-world measurement noise (in $L_2$ relative error for all and error rate for Darcy inverse).

| | Steps $(N)$ | Darcy Flow | | Poisson | | Helmholtz | |
|---|---|---|---|---|---|---|---|
| | | Forward | Inverse | Forward | Inverse | Forward | Inverse |
| FNO | – | 51.78% | 6839.16% | 55.84% | 1.083e07% | 78.31% | 2.19e05% |
| PINO | – | 45.53% | 12461.1% | 61.04% | 8.56e05% | 62.19% | 6.84e05% |
| DiffusionPDE | 2000 | 36.4% | 68.1% | 39.61% | 127.49% | 22.58% | 113.17% |
| FunDPS | 200 | 9.18% | 46.6% | 13.81% | 806.45% | 14.33% | 467.63% |
| PRISMA | 20 | **8.22%** | **20.3%** | **12.71%** | **35.49%** | **12.04%** | **59.99%** |

Table 16: Comparison of different models on three PDE problems with 30% pixels having Unit Gaussian noise corruption, simulating real-world measurement noise (in $L_2$ relative error for all and error rate for Darcy inverse).

| | Steps $(N)$ | Darcy Flow | | Poisson | | Helmholtz | |
|---|---|---|---|---|---|---|---|
| | | Forward | Inverse | Forward | Inverse | Forward | Inverse |
| FNO | – | 11.21% | 548.81% | 14.32% | 836.31% | 48.2% | 773.04% |
| PINO | – | 10.06% | 936.3% | 11.49% | 682.01% | 40.23% | 531.61% |
| DiffusionPDE | 2000 | 35.21% | 63.26% | 25.78% | 117.36% | 22.14% | 108.04% |
| FunDPS | 200 | 8.53% | 42.5% | 11.28% | 796.22% | 13.15% | 384.78% |
| PRISMA | 20 | **6.23%** | **17.97%** | **11.22%** | **32.16%** | **9.16%** | **54.43%** |

## I  QUALITATIVE RESULTS

Figure 13 provides a qualitative comparison of the solutions generated by PRISMA and the baseline models under the challenging noisy observation setting. Across both problems: Poisson and Darcy Flow, PRISMA's predictions are visually faithful to the ground truth solutions. The corresponding

Table 17: Comparison of different models on three PDE problems with 10% pixels having Unit Gaussian noise corruption, simulating real-world measurement noise (in $L_2$ relative error for all and error rate for Darcy inverse).

| | Steps ($N$) | Darcy Flow | | Poisson | | Helmholtz | |
|---|---|---|---|---|---|---|---|
| | | Forward | Inverse | Forward | Inverse | Forward | Inverse |
| **FNO** | – | 7.68% | 395.41% | 9.36% | 477.67% | 23.56% | 511.91% |
| **PINO** | – | 5.35% | 545.59% | **6.52%** | 461.09% | 17.9% | 413.07% |
| **DiffusionPDE** | 2000 | 33.12% | 53.12% | 17.51% | 107.14% | 19.99% | 107.72% |
| **FunDPS** | 200 | 8.5% | 31.34% | 10.63% | 598.7% | 10.84% | 239.65% |
| **PRISMA** | 20 | **3.77%** | **12.34%** | 7.3% | **27.12%** | **8.05%** | **43.56%** |

error maps are consistently darker, and the reported error values are significantly lower compared to the baselines. In contrast, solutions from FunDPS often appear corrupted by noise, while those from DiffusionPDE can be blurry or inaccurate, particularly in complex inverse problems. These visualizations provide strong qualitative evidence of PRISMA's robustness and superior performance when dealing with imperfect data. Figures 14, 15, 16 presents qualitative comparison of the solutions generated by PRISMA and the baseline models under Full observation setting. Figure 17 compare PRISMA predictions against the baseline models on Navier-Stokes Equation under Sparse observation settings.

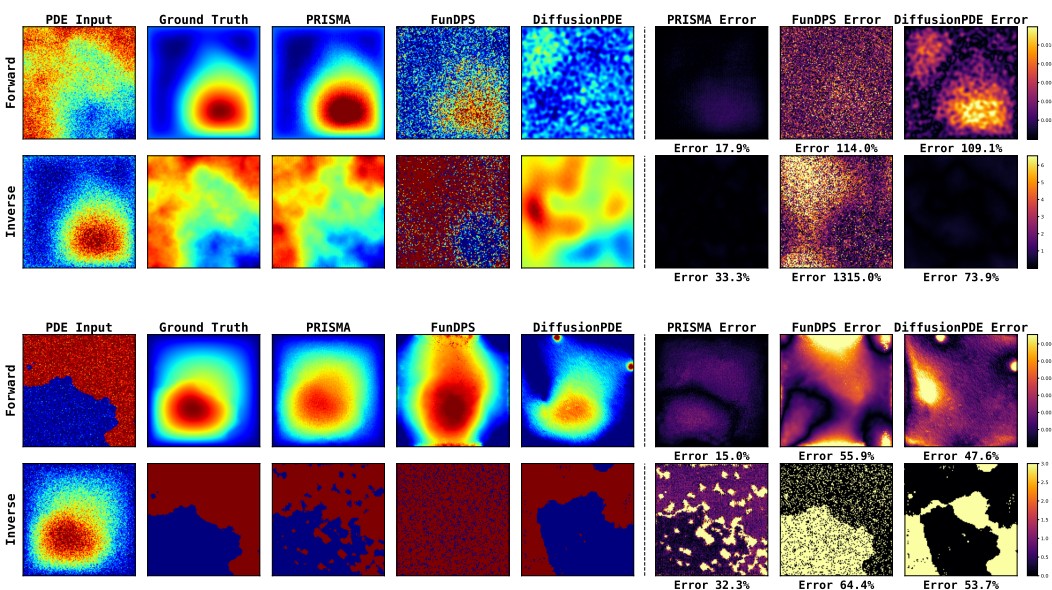

Figure 13: Poisson (*top*) and Darcy (*bottom*) Equations under Noisy Observation

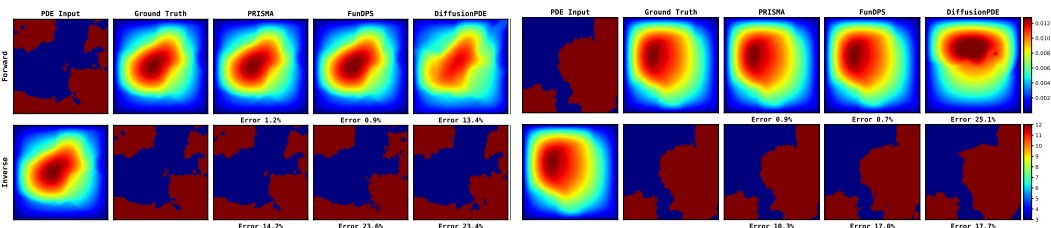

Figure 14: Prediction (*for two samples*) Visualization on Darcy Equation under Full Observation

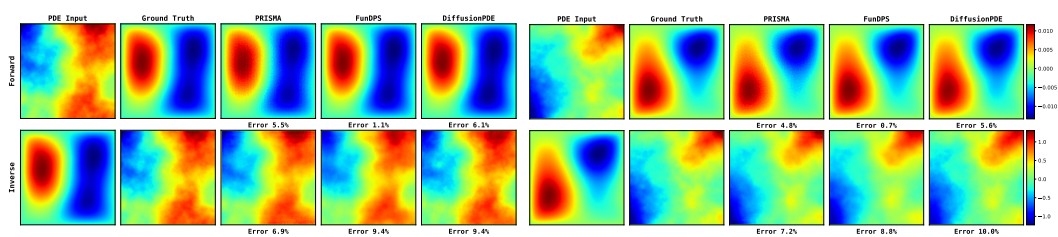

Figure 15: Prediction (*for two samples*) Visualization on Helmholtz Equation under Full Observation

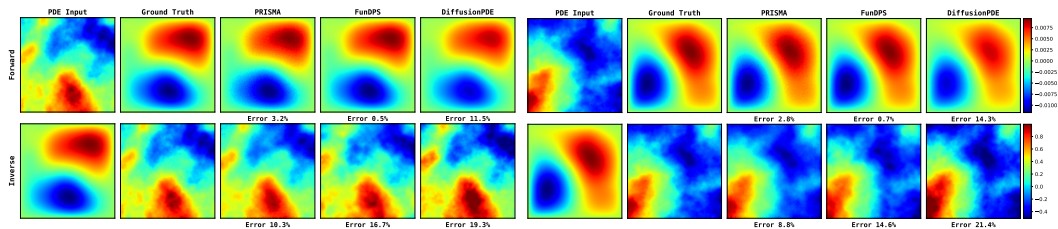

Figure 16: Prediction (*for two samples*) Visualization on Poisson Equation under Full Observation

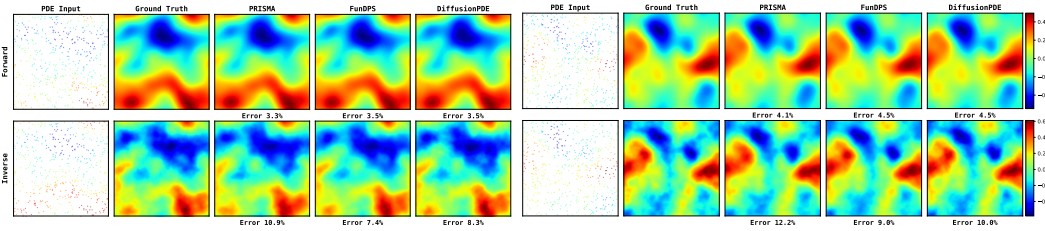

Figure 17: Prediction (*for two samples*) Visualization on Navier-Stokes (non bounded) Equation under Sparse Observation

