# OpenReview forum: "Beyond Loss Guidance: Using PDE Residuals as Spectral Attention in Diffusion Neural Operators"
_ICLR.cc/2026/Conference — Submitted to ICLR 2026_

### Official Review · Reviewer_hiiL · 2025-10-28

**Soundness:** 3
**Presentation:** 2
**Contribution:** 2
**Rating:** 4
**Confidence:** 4

**Summary:**

This paper proposes a diffusion-based PDE solver that can significantly accelerate the problem solving process. The new design is that the author introduces a conditional diffusion neural operator that embeds PDE residuals directly into the model’s architecture via attention mechanisms in the spectral domain, enabling gradient-free inference. The experimental results show PRISMA is at-par or better in accuracy compared to previous methods across five benchmark PDEs especially with noisy observations, while using 10x to 100x fewer denoising steps, leading to 15x to 250x faster inference.

**Strengths:**

(1) The novelty is somewhat good. Unified Framework for Diverse PDE Tasks: PRISMA’s conditional design, enabled by input masks unifies forward and inverse PDE solving under a single model, supporting full, sparse, and noisy observation regimes. Unlike baselines (e.g., FNO, PINO) that require task-specific models or inference pipelines, PRISMA seamlessly adapts to different problem settings (e.g., sparse Darcy flow inverse problems, noisy Navier-Stokes forward problems) without reconfiguration. This versatility is a significant advance for real-world applications where observation quality varies.

(2) The experimental results. The paper demonstrates compelling empirical performance across five benchmark PDEs (Darcy Flow, Poisson, Helmholtz, Navier-Stokes with/without BCs). PRISMA achieves 15x–250x faster inference (0.18–0.8 seconds per sample) compared to diffusion-based baselines (e.g., DiffusionPDE: 213s, FunDPS: 11.8s) by using only 20–50 denoising steps (vs. 200–2000 steps for baselines). Crucially, this speedup does not compromise accuracy: PRISMA outperforms baselines in noisy settings (e.g., Darcy Flow forward error: 12.28% vs. FunDPS’s 55.09% and DiffusionPDE’s 49.18%) and matches top performers (e.g., PINO, FunDPS) in full/sparse observations.

**Weaknesses:**

(1) Limited Discussion of SRA Block Mechanics: While the SRA block is core to PRISMA’s success, the paper provides insufficient detail on its internal workings. For example:
The calculation of the compatibility score
S^l(k) (complex inner product of Fourier-transformed features and residuals) is described, but the intuition for why spectral-domain attention outperforms spatial-domain methods (e.g., residual concatenation) is underdeveloped.
The MLP that learns g_res (guidance strength) is not specified (e.g., architecture, input features beyond r_avg and c_σ), making it hard to replicate or extend.
The paper does not explain how SRA handles frequency modes with conflicting residual signals (e.g., high-frequency noise vs. low-frequency physical signals).


(2) Spatio-Temporal and Irregular Mesh Limitations: PRISMA is evaluated exclusively on static (time-independent) PDEs with regular grid inputs. The authors acknowledge future work will extend to spatio-temporal problems and irregular meshes, but the current limitation narrows PRISMA’s applicability. Many real-world PDEs (e.g., time-dependent Navier-Stokes, heat equation on complex geometries) require these capabilities, and the paper provides no insight into how PRISMA’s architecture might adapt (e.g., integrating temporal attention, handling unstructured grids). Are there any insights for this problem.

(3) Scalability to High Resolution: The paper evaluates PRISMA on 64×64 and 128×128 grids, but does not address scalability to larger resolutions (e.g., 256×256, 512×512). Diffusion models and neural operators often face computational bottlenecks at high resolution (e.g., Fourier transform costs, memory usage). The authors should clarify whether PRISMA’s 64M parameter count (similar to baselines) remains feasible at higher resolutions, or if modifications (e.g., sparse Fourier transforms) are needed.

**Questions:**

Please see the weakness.

---

> ### Author Response · Authors · 2025-11-19
> **Author Response to Reviewer hiiL**
>
> Thank you for the comments. Please find our responses to individual comments below.
>
> **Comment 1: Limited Discussion of SRA Block Mechanics: While the SRA block is core to PRISMA’s success, the paper provides insufficient detail on its internal workings. For example: The calculation of the compatibility score S^l(k) (complex inner product of Fourier-transformed features and residuals) is described, but the intuition for why spectral-domain attention outperforms spatial-domain methods (e.g., residual concatenation) is underdeveloped.**
>
> > We apologize for the concise description of PRISMA mechanics due to space constraints and are happy to elaborate on the inner workings of PRISMA.
>
> > There are two key intuitions behind our use of attention in the spectral domain as opposed to the spatial domain. First, spectral methods are great at capturing both local and global signals, which is one of the reasons for using UNO as a backbone in PRISMA. To attend to PDE residuals, we need to translate local (and often subtle) PDE violations globally to the entire solution field: a task where spectral methods are a natural fit. In contrast, spatial operations are susceptible to blurring or smoothening local fluctuations in PDE residuals. Second, spectral attention allows precise correction of the solution field based on frequency mode-wise control of PDE residuals. Specifically, we can amplify or suppress the importance of different frequencies of the PDE residual by operating in the spectral space. This is especially important because PDE residual fields are known to show varying sensitivities to different frequencies during loss optimization (see previous works relating failure modes of PINNs to high frequency components of PDE residuals: [1][2]).
>
> > We empirically demonstrate the value of using residuals in the spectral domain as opposed to concatenating residuals in the spatial path in Table 4. We also provide new results by ablating different components of our SRA block in Table 13 of Appendix, to further analyze SRA’s design and isolate the effects of the gate, phase awareness, and skip vs. multiplicative gating.
>
> > [1] Arka Daw, Jie Bu, Sifan Wang, Paris Perdikaris, and Anuj Karpatne. 2023. Mitigating propagation failures in physics-informed neural networks using retain-resample-release (r3) sampling. In Proceedings of the 40th International Conference on Machine Learning (ICML'23), Vol. 202. JMLR.org, Article 288, 7264–7302.
>
> > [2] Krishnapriyan, Aditi, Amir Gholami, Shandian Zhe, Robert Kirby, and Michael W. Mahoney. "Characterizing possible failure modes in physics-informed neural networks." Advances in neural information processing systems 34 (2021): 26548-26560.
>
>
> **Comment 2: The MLP that learns g_res (guidance strength) is not specified (e.g., architecture, input features beyond r_avg and c_σ), making it hard to replicate or extend.**
>
> > The exact implementation of the guidance strength gate has been provided in our submitted code (module ResidualGate) allowing full reproducibility of our framework. We describe it below and have also moved this discussion to the Appendix.
>
> > Every SRA block uses a small two-layer MLP to predict the scalar guidance weight $g_{\text{res}}\in[0,1]$. The inputs are (i) the diffusion/timestep embedding $c_\\sigma \text{(of length E, e.g., E}{=}256)$ and (ii) the spatial mean of the residual $r_{\text{avg}}$ (a scalar). We concatenate them to a $B\times(E{+}1)$ tensor where B is batch size and pass it through $Linear(E{+}1\rightarrow E) → ReLU → Linear(E\rightarrow 1) → Sigmoid$, yielding a $B\times1$ output used for skip-connected gating in SRA. Every SRA block has its own small MLP.
>
> **Comment 3: The paper does not explain how SRA handles frequency modes with conflicting residual signals (e.g., high-frequency noise vs. low-frequency physical signals).**
>
> > SRA resolves conflicting residual frequency signals through three mechanisms. First, the phase-aware complex inner product down-weights incoherent high-frequency noise while reinforcing frequency modes aligned with the physical residual. Second, the learned per-mode spectral gain weights $w_{\text{gain}}(\mathbf{k})$ calibrate which frequencies to attend. Third, the scalar gate $g_{\text{res}}$ modulates the influence of the residual across the denoising trajectory, reducing the effect of noisy residuals at early or unstable steps. Following the UNO/FNO design, SRA operates only on the retained Fourier modes after spectral truncation, ensuring that we do not attend to modes that are eventually truncated by FNO right after the SRA.

---

> ### Author Response · Authors · 2025-11-19
> **Author Response to Reviewer hiiL (contd.)**
>
> **Comment 4 Spatio-Temporal and Irregular Mesh Limitations: PRISMA is evaluated exclusively on static (time-independent) PDEs with regular grid inputs. The authors acknowledge future work will extend to spatio-temporal problems and irregular meshes, but the current limitation narrows PRISMA’s applicability. Many real-world PDEs (e.g., time-dependent Navier-Stokes, heat equation on complex geometries) require these capabilities, and the paper provides no insight into how PRISMA’s architecture might adapt (e.g., integrating temporal attention, handling unstructured grids). Are there any insights for this problem.**
>
> > We agree that applying our work on more challenging problems involving spatio-temporal data and irregular meshes is a valuable contribution. However, we chose the exact same suite of benchmark PDE experiments as used in recent works (DiffusionPDE and FunDPS) to maintain consistency and fairness in comparison of results. Note that a practical challenge in evaluating baseline methods such as DiffusionPDE on new datasets is their high sensitivity to hyper-parameter settings and issues with reproducibility that have been documented in previous works. To ensure a fair comparison, we used reported checkpoints of baselines on known datasets without extending them, which can certainly be the scope of future works.
>
> > To show the generalizability of PRISMA to spatio-temporal problems, we performed new experiments comparing PRISMA with DiffusionPDE on the 1D-Burgers equation where the goal is to predict the full-time solution field given sparse spatial observations across all timesteps. Following the experiment setup of DiffusionPDE, we randomly selected 5 out of 128 spatial sensors for creating sparse inputs. We reproduced DiffusionPDE’s results using their publicly released checkpoint and default hyperparameters. The relative errors on this experiment are summarized in the Table below.
>
> > ### Relative Error (%) on 1D Burgers equation
> | **Method**       | **Relative Error (%)** | **Steps** |
> |------------------|------------------------|-----------|
> | **DiffusionPDE** | 10.39%                 | 2000      |
> | **PRISMA**       | 9.33%                  | 20        |
>
> > We can see that PRISMA achieves a slightly better accuracy than DiffusionPDE while reducing inference from 2000 guidance steps to 20 feed-forward steps. We have added more details and visualizations in Appendix F of the paper. For more general spatio-temporal problems involving 2D and 3D spatial domains, one way we can extend PRISMA is by representing multiple time slices as stacked channels of inputs and outputs, allowing the spectral blocks to learn cross-time interactions implicitly.
>
> > PRISMA can also be extended to irregular geometries by adopting a latent-space diffusion formulation where we first encode inputs and outputs to a regular latent grid, learn diffusion neural operators, and decode the latent space back to the original space of inputs and outputs. Since our SRA block is backbone-agnostic, it can also be plugged with any diffusion backbone. We will discuss these points in the future works section.

---

> ### Author Response · Authors · 2025-11-19
> **Author Response to Reviewer hiiL (contd. 2)**
>
> **Comment 5 Scalability to High Resolution: The paper evaluates PRISMA on 64×64 and 128×128 grids, but does not address scalability to larger resolutions (e.g., 256×256, 512×512). Diffusion models and neural operators often face computational bottlenecks at high resolution (e.g., Fourier transform costs, memory usage). The authors should clarify whether PRISMA’s 64M parameter count (similar to baselines) remains feasible at higher resolutions, or if modifications (e.g., sparse Fourier transforms) are needed.**
>
> > Thank you for raising this question on scalability. While our datasets provide ground truth only up to 128×128, we added a super-resolution experiment to assess PRISMA’s scalability to higher resolution. We applied our model trained on 64×64, without retraining, to 128×128, 256×256, and 512×512 inputs generated via interpolation. We have added visualizations of this experiment in Appendix G where we have also included a Table on inference time and GPU memory trends of PRISMA by changing resolution. The results show that 256×256 is feasible on a single A100 while 512x512 begins to strain single-GPU memory. For resolutions beyond this range, we recommend leveraging scaling strategies used in spectral and diffusion models such as mixed precision, latent diffusion models, or sparse FFTs with fewer modes.
>
> > ### Super-resolution Inference Time and Memory Tracking (Darcy)
> | **Resolution** | **Avg. Batch Time (s)** | **Avg. Per-Sample Time (s)** | **Avg. Peak GPU Mem (GB)** | **Avg. Reserved GPU Mem (GB)** |
> |----------------|--------------------------|-------------------------------|------------------------------|---------------------------------|
> | **64×64**      | 12.140 ± 0.945           | 0.242 ± 0.018                 | 2.781 ± 0.572                | 4.117 ± 0.090                   |
> | **128×128**    | 20.036 ± 0.326           | 0.400 ± 0.006                 | 6.421 ± 0.573                | 15.383 ± 0.804                  |
> | **256×256**    | 60.892 ± 0.580           | 1.217 ± 0.011                 | 20.983 ± 0.573               | 62.602 ± 0.804                  |
> | **512×512**    | 242.489 ± 0.423          | 4.849 ± 0.008                 | 79.194 ± 0.573               | 107.114 ± 0.399                 |

---

### Official Review · Reviewer_851Q · 2025-10-29

**Soundness:** 2
**Presentation:** 2
**Contribution:** 1
**Rating:** 2
**Confidence:** 4

**Summary:**

This paper introduces a PDE solver framework called PRISMA. This framework is a diffusion model which encodes the physical information into the structure of the denoiser, but not reinforced through physical loss. The embedding of physical information is through a mechanism called Spectral Residual Attention (SRA), which includes the transformation of physical residual into frequency domain, and then attention with the observed information also in frequency domain. The performance of their proposed method is better than other models on PDE problems with 100% unit Gaussian noise corruption.

**Strengths:**

- (Originality) This work introduces a new framework that can incorporate physical information other than direct calculation and backpropagation of physical loss, which seems to reduce the training time and inference steps for diffusion-based frameworks.
- (Clarity) Besides some minor issues with mathematical symbols (see weaknesses part), the general presentation of this work is easy to follow.

**Weaknesses:**

- (Wrong Physical Residual Calculation of NS) One of the most serious problems with this paper is that, the calculation of physical residual for nonbounded NS equations, which was adopted from DiffusionPDE, is actually wrong. The vorticity $\vec{\omega}(x,y)=\vec{\nabla} \times \vec{v}(x, y)$ is an (axial-)vector which only has $z$ component and is a function of $x, y$. Therefore, its zero divergency, $\vec{\nabla} \cdot \vec{\omega}(x,y) = \frac{\partial \omega}{\partial z} = 0$ cannot be regarded as a meaningful physical residual. The same thing happens to bounded NS. In that case, only the magnitude of $\vec{v}(x, y)$ is recorded, and one cannot take divergence on a magnitude field. This serious problem would make all the arguments for physical embedding useless, as a comparison of performance without physical residual and with a wrong physical residual does not make any sense.
- (Problems Tested) This paper compares the performance of models on a rather rare case of observations with 100% Gaussian noise. In reality if one observation is with 100% noise, it is considered a failed observation. One more serious problem is about the physical residual calculated on observation with 100% noise. This noise is high frequency and would hurt the physical residual calculated with finite difference. One cannot get convincing physical information from a physical residual that is not reliable.
- (Performance and Baseline) On more common full observation and partial observation, the performance of this model is inferior to other models like PINO and FunDPS, as reported in the appendices. The baseline of DiffusionPDE on full observation is actually available in their paper, and is much better than the results reported in the Table 6 of this paper.
- (Mathematical Symbol) In equation (1), $\mathcal{F}$ stands for the operator of PDE, but in equation (2), $\mathcal{F}$ stands for the Fourier transform operator. This reuse of symbol hurts the clarity of this paper.

**Questions:**

- To show that the proposed method works for NS, maybe the authors can try out with vector form of velocity field. The zero divergency of velocity vector field can be implemented as: $\vec{\nabla}\cdot \vec{v}(x, y) = \frac{\partial v_x}{\partial x} + \frac{\partial v_y}{\partial y} = 0$.

---

> ### Author Response · Authors · 2025-11-22
> **Author Response to Reviewer 851Q**
>
> Thank you for the comments. Please find our responses to individual comments below.
>
> **Comment 1: (Wrong Physical Residual Calculation of NS) One of the most serious problems with this paper is that, the calculation of physical residual for nonbounded NS equations, which was adopted from DiffusionPDE, is actually wrong. The vorticity  is an (axial-)vector which only has  component and is a function of . Therefore, its zero divergency,  cannot be regarded as a meaningful physical residual. The same thing happens to bounded NS. In that case, only the magnitude of  is recorded, and one cannot take divergence on a magnitude field. This serious problem would make all the arguments for physical embedding useless, as a comparison of performance without physical residual and with a wrong physical residual does not make any sense.**
>
>
> > Thank you for raising this important point. We fully agree that the computation of the PDE residual in the Navier–Stokes (NS) dataset, which was introduced by DiffusionPDE and has since become a standard benchmark in the field of diffusion-based PDE solvers such as FunDPS, is physically incorrect. Specifically, divergence $\nabla \cdot \mathbf{v}$ is only defined for a *vector* field $\mathbf{v}$, and cannot be applied to a scalar field $f(x,y)$ such as vorticity or a single component of velocity. In DiffusionPDE’s codebase, the PDE loss is actually implemented as the sum of first order derivatives of $f$, $\frac{\partial f}{\partial x} + \frac{\partial f}{\partial y}$, which is neither divergence nor a meaningful physics residual. We would also like to clarify the following additional issues regarding DiffusionPDE’s implementation of the PDE residual loss:
>
> > 1. For the **non-bounded case**, DiffusionPDE considers the sum of first-order derivatives of vorticity $\omega$ that has no physical justification and is not generally zero. Also, even if one reconstructs the velocity field $\mathbf{v}$ from $\omega$ via stream function, the incompressibility constraint $\nabla \cdot \mathbf{v} = 0$ cannot be used as a residual because it is trivially satisfied by the stream function formulation. Hence, the only valid residual is the momentum equation, as pointed by the reviewer. However, since the original setup involves predicting the final vorticity at $T = 1$ sec from the initial vorticity at $T = 0$, computing the temporal derivative numerically is inaccurate over such a long interval. We nevertheless implemented this during the rebuttal and observed trends consistent with the rest of our experiments.
>
> > 2. For the **bounded case**, DiffusionPDE’s dataset is set up such that only a single channel of velocity is stored at every time-step, but it is not specified whether it is the x-component, the y-component, or the magnitude (this issue has been raised in their public Github repo but has not been resolved yet). With only a single velocity channel, it is not possible to compute the momentum equation residual or the incompressibility residual.
>
> >We sincerely appreciate the reviewer’s feedback in identifying this issue that has been inadvertently propagating in the diffusion-based PDE community and we believe fixing this issue will help the field move forward toward physically grounded evaluation frameworks. At the same time, we respectfully argue that this dataset issue is external to our method and does not affect the contributions of our work. Our key finding, that PRISMA is 15×–250× faster inference while achieving competitive accuracy, still holds true on the four other PDE benchmarks: Darcy, Poisson, Helmholtz, and Burgers, which has well-defined and physically correct PDE residuals.
>
> >In response to this comment, we have removed all results on both NS datasets from the revised manuscript and the rebuttal comments. As requested by the reviewer, we are currently running new experiments on the non-bounded NS case using the momentum equation residual with a short time interval of 0.1 seconds to numerically compute time derivative of vorticity. We are happy to share these results as soon as the runs are complete.

---

> ### Author Response · Authors · 2025-11-22
> **Author Response to Reviewer 851Q (contd.)**
>
> **Comment 2: (Problems Tested) This paper compares the performance of models on a rather rare case of observations with 100% Gaussian noise. In reality if one observation is with 100% noise, it is considered a failed observation. One more serious problem is about the physical residual calculated on observation with 100% noise.
> This noise is high frequency and would hurt the physical residual calculated with finite difference. One cannot get convincing physical information from a physical residual that is not reliable.**
>
>
> > Response: We would like to clarify that by “100% Gaussian noise”, we mean that we *add* unit Gaussian noise to the observation at every pixel (reflecting measurement noise) rather than replacing the original signal with complete noise (which would reflect the extreme case of absolute sensor failure). We now also include a noise sweep at 10/30/50/90% pixel corruptions in the Appendix (Tables 15-18) and observe consistent trends across methods. This setting corresponds to common real-world sensor degradations where the signal is present but heavily perturbed, rather than completely missing. Moreover, in Figure 6 we also analyze model performance in comparison to baselines on varying intensities of Gaussian noise ($\sigma$).
>
> >We agree that residuals computed on heavily corrupted observations can be unreliable. PRISMA addresses this via the scalar guidance weight $g_{\text{res}}\in[0,1]$, which is conditioned on the spatial residual. This gate attenuates or ignores the residual pathway when it appears inconsistent/high-variance, effectively defaulting toward observation-only conditioning.
>
>
> **Comment 3: A. (Performance and Baseline) On more common full observation and partial observation, the performance of this model is inferior to other models like PINO and FunDPS, as reported in the appendices.**
>
> >We agree that FNO/PINO are strong choices for fully observed inputs and have low inference latency, and that FunDPS is also very accurate in some sparse cases. Our goal, however, is to develop a **single unified model** that handles full/partial/noisy observations *without per-instance test-time optimization loops**, while keeping the sampling cost low (see Table 1). Under full observations (Table 6), PRISMA remains competitive in accuracy across all PDEs and **achieves the second-best average rank (2.83)** when compared with all baselines (including FNO/PINO) while using 20-50 sampling steps.
>
> >There are a few tasks where PRISMA’s error is slightly higher than the best baseline, but it offers a significantly better accuracy-latency tradeoff. Relative to the sparse strongest baseline (FunDPS-500), PRISMA is within small relative % errors while being 15x to 65x faster per sample:
>
> >### Performance and Latency Comparison
> | **Model (steps)** | **Darcy For Δ** | **Darcy Inv Δ** | **NS For Δ** | **NS Inv Δ** | **Per-sample latency** | **Speedup vs FunDPS-500** |
> |-------------------|------------------|------------------|--------------|--------------|--------------------------|-----------------------------|
> | **PRISMA-20**     | +0.64            | +1.62            | +0.58        | +3.25        | 0.18 s                   | ~65×                        |
> | **PRISMA-50**     | +0.62            | +1.62            | +0.53        | +3.02        | 0.80 s                   | ~15×                        |
> (Δ = relative percentage gap to FunDPS-500; lower is better.)
>
> >Importantly, PRISMA maintains competitive accuracy with significantly lower inference cost, requiring a significantly small number of steps than baselines.

---

> ### Author Response · Authors · 2025-11-22
> **Author Response to Reviewer 851Q (contd. 2)**
>
> **Comment 4: The baseline of DiffusionPDE on full observation is actually available in their paper, and is much better than the results reported in the Table 6 of this paper.**
>
> > We reproduced DiffusionPDE/FunDPS using their exact public configs, released checkpoints, per-equation default guidance weights, and suggested step counts for all full/sparse/noisy settings. Given their high sensitivity to seed/config setups (see Appendix D) that is also reported in prior works (see Appendix I of FunDPS paper), we chose not to perform any hyper-parameter tuning of baselines to be faithful in reproducing their original works. We quote prior numbers from their tables wherever we were able to reproduce their results within a reasonable range. Otherwise, we re-ran using their default configurations (on multiple seeds) and reported the observed values (these revised numbers are highlighted using asterisk in Table 6). Below we report our reproduced numbers across three seeds and our paper reports the reproduced (not paper-quoted) results for fairness and transparency.
>
> > ### Diffusion PDE-  Full Observations (Reproduced Results with Different Seeds)
> | **Equation** | **Setting** | **DiffusionPDE Paper** | **Reproduced (Seed 4982)** | **Reproduced (Seed 7487)** | **Reproduced (Seed 92)** |
> |--------------|-------------|-------------------------|-----------------------------|-----------------------------|---------------------------|
> | **Darcy Flow** | forward | 2.20% | 3.91% | 3.01% | 2.91% |
> |              | inverse | 2.00% | 15.07% | 14.50% | 13.02% |
> | **Poisson**  | forward | 2.70% | 15.33% | 15.89% | 15.27% |
> |              | inverse | 9.80% | 25.06% | 27.20% | 21.21% |
> | **Helmholtz** | forward | 2.30% | 17.87% | 19.76% | 10.90% |
> |              | inverse | 4.00% | 18.40% | 18.96% | 18.97% |
> | **Non-bounded Navier–Stokes** | forward | 6.10% | 3.68% | 3.76% | 2.40% |
> |                               | inverse | 8.60% | 9.61% | 9.59% | 8.40% |
> | **Bounded Navier–Stokes** | forward | 1.70% | 5.73% | 5.20% | 2.80% |
> |                           | inverse | 1.40% | 3.76% | 6.60% | 1.95% |
>
> **Comment 5:  (Mathematical Symbol) In equation (1),  stands for the operator of PDE, but in equation (2),  stands for the Fourier transform operator. This reuse of symbol hurts the clarity of this paper.**
>
> > Response: Thank you for pointing this out. We have fixed this now in the manuscript: the PDE operator is denoted by $\mathcal{N}$ while the Fourier transform is denoted by $\mathcal{F}$.

---

> ### Author Response · Authors · 2025-11-27
> **Author Response to Reviewer 851Q (contd. 3 - follow-up for NS non-bounded)**
>
> > We re-ran the non-bounded NS experiments as suggested: we now predict the next time step (rather than the final frame) with $\Delta t=0.1$. We reconstruct velocity from vorticity via the streamfunction $(\Delta\psi=-\omega,\;\mathbf{v}=(\partial_y\psi,-\partial_x\psi))$, and use the momentum (vorticity-transport) equation residual for guidance. The updated results are:
> >### Navier–Stokes Performance Summary
> | **Model**       | **Steps** | **Full NS (Fwd)** | **Full NS (Inv)** | **Noisy NS (Fwd)** | **Noisy NS (Inv)** | **Sparse NS (Fwd)** | **Sparse NS (Inv)** | **Avg. Rank** |
> |-----------------|-----------|--------------------|--------------------|---------------------|---------------------|----------------------|----------------------|----------------|
> | **FunDPS**      | 200       | 4.00%              | 1.68%              | **31.55%**              | 38.44%              | 11.89%               | 9.16%                | 2.5            |
> | **DiffusionPDE**| 2000      | **0.60%**          | 1.50%              | 109%                | 46.00%              | 7.87%                | **8.00%**            | 2              |
> | **PRISMA**      | 20        | 1.51%              | **1.30%**          | 36.80%          | **37.50%**          | **7.15%**            | **8.00%**            | **1.33**       |
>
> >We observe that DiffusionPDE performs best in full observation settings, but is not robust to noise. PRISMA stays competitive in noisy and sparse settings and achieves the best overall rank with far fewer steps.

---

> > ### Comment · Reviewer_851Q · 2025-11-28
> >
> > I would like to thank the authors for their clarifications and additional experiments. Some of the experimental results still look very skeptical to me. For example, in Tables 14-16, the error rates of Darcy Flow Inverse problems should be at most 100%, so I don't understand why there are results as high as 8974.61%. I encourage the authors to review their results carefully before they submit them.

---

> ### Author Response · Authors · 2025-12-03
> **Author Response to Reviewer 851Q**
>
> Thank you for reading our rebuttal. Here are our individual responses to the follow-up comments.
>
> **Comment: Some of the experimental results still look very sceptical to me. For example, In Tables 14-16, the error rates of Darcy Flow Inverse problems should be at most 100%, so I don't understand why there are results as high as 8974.61%. I encourage the authors to review their results carefully before they submit them.**
> > Thank you for pointing this out. This discrepancy arose because we reported relative L2 (%) for the Darcy Flow inverse tasks for PINO/FNO instead of the coefficient error rate. We have corrected all Tables to report coefficient error rate for Darcy inverse to maintain consistency with previous works.

---

### Official Review · Reviewer_H4cn · 2025-10-30

**Soundness:** 2
**Presentation:** 3
**Contribution:** 2
**Rating:** 4
**Confidence:** 4

**Summary:**

The authors propose PRISMA (PDE Residual Informed Spectral Modulation with Attention). They modify the global spectral path of a UNO, via a residual guided attention mechanism; which depends on the phase alignment of said PDE residuals, as well as a gate dependent on the diffusion step/noise level. This speeds up the inference time of the models by a 15-250x, while maintaining or surpassing their compared models.

**Strengths:**

### Novel PDE-informed guidance mechanism
Using complex-valued attention in Fourier space to modulate frequencies based on PDE residuals is genuinely novel. The authors established via gating and normalization a well rounded method to perform this task.

### Impressive speedup of the inference (for diffusion models)
The 20-step inference while achieving comparative accuracy in some tasks is impressive. Having a speedup of 15-250x is the strongest contribution of this work. This brings the neural PDE solvers a step closer to being practical in real-time applications.

### Unified framework (forward-inverse problems)
Training on both forward and inverse problem simultaneously with task-specific probabilities is elegant.

### Provided code
This made the review process easier to understand how the experiments were setup/run.

**Weaknesses:**

### Method is only decent on full observations, compared to other models (Table 6).
In this case PINO and the other models seem to outperform the proposed method (in speed and accuracy)

### Noise robustness claims (Table 3).
Overall this table shows that diffusion models are better suited for noisy data. You should compare PINO/FNO trained with the same data augmentation (noise injection during training). The current comparison falls flat, as this setting would be out-of-distribution data for PINO/FNO, while it is in-distribution for most diffusion methods.
A similar claim can be made for Table 7 results (sparse observations).

### More ablation runs for Table 4.
Without confidence intervals, the differences in Table 4 are questionable. Some tasks show minimal improvment or even worse performance without PDE residuals. The paper should report mean and std over multiple seeds (code only shows seed:33).

### Claim of multi-scale guidance is somewhat misleading
The paper claims to provide "multi-scale guidance at every layer of UNO", but the implementation uses interpolation by downsampling to the same resolution to match the feature map spatial dimensions. Unlike hierarchical feature extractors, this method appears to not learn different representations of PDE residuals at different scales (simple resize of the same signal). This means the multi-scale is rather a multi-resolution.

### The caption of Figure 4 appears to be incorrect
Caption does not match the figure content (mentions "inference time vs accuracy", but this is not shown)

**Questions:**

- Is there any intuition why FUNDPS outperforms the other methods in the sparse observation case? (Table 7)

- Did you also try out to integrate magnitude alignment into the attention? Phase alignment and using w_gain appears to perform well, I wonder if magnitude information could be used to omit the w_gain term.

- Can you provide results where PINO and other baselines are trained with the similar noise augmentation schedule?

- Can you report mean and std for +5 random seeds? (Table 4)

- Have you ablated using magnitude-only vs. phase-only alignment in frequency domain?

- Why is complex-valued attention necessary, compared to for example attention in the spatial path?

---

> ### Author Response · Authors · 2025-11-19
> **Author Response to Reviewer H4cn**
>
> Thank you for the comments. Please find our responses to individual comments below.
>
> **Comment 1: Method is only decent on full observations, compared to other models (Table 6). In this case PINO and the other models seem to outperform the proposed method (in speed and accuracy)**
>
> > We agree that FNO/PINO are strong choices for fully observed inputs and have lower inference latency. Our goal, however, is to develop a single unified model that also remains robust in noisy and sparse settings without any test-time optimization, that are outside the scope of PINO and related operator-learning methods as summarized in our related-works comparison (see Table 1). Under full observations (Table 6), PRISMA remains competitive in accuracy across all PDEs and **achieves the best average rank** of 2.4 when compared with all baselines (including FNO/PINO). Importantly, PRISMA maintains competitive accuracy with significantly lower inference cost, requiring a significantly small number of steps than baselines.
>
> **Comment 2: Noise robustness claims (Table 3). Overall this table shows that diffusion models are better suited for noisy data. You should compare PINO/FNO trained with the same data augmentation (noise injection during training). The current comparison falls flat, as this setting would be out-of-distribution data for PINO/FNO, while it is in-distribution for most diffusion methods. A similar claim can be made for Table 7 results (sparse observations). Can you provide results where PINO and other baselines are trained with the similar noise augmentation schedule?**
>
> > We would like to clarify two important details.
> > 1. None of the diffusion models used in our work (PRISMA, DiffusionPDE, and FunDPS) were trained with any form of data augmentation relating to noise injection in the input observations. We only use noisy inputs during inference, making the training of all models agnostic to observation noise and the test setting to be out-of-distribution for all methods including PINO and FNO.
>
> > 2. There is a distinction between noise in the conditioning inputs (or “observation noise”) and “diffusion noise” that is inherent to the forward and reverse diffusion processes applied to the target field, to learn a denoiser during training or to apply the trained denoiser on starting noise during inference. Concretely, a clean target u is perturbed as $u_t=\alpha_t u+\sigma_t\epsilon$ and the model learns to predict $\epsilon$ conditioned on clean observations/residuals. This is fundamentally different from the observation noise on conditioning inputs that we introduce only during inference.
>
> > Regarding sparse observations, we would like to make a similar clarifying statement that PRISMA training is agnostic to sparsity levels encountered during inference. We train PRISMA on a unified set of diverse sparsity levels in inputs rather than a fixed sparsity level or missingness pattern. More crucially, we do not assume knowledge of or tune our model to the type of masks or sparsity levels encountered during inference, which still represent out-of-distribution testing scenarios for all methods.
>
> > Also, training FNO/PINO with a noise schedule similar to the forward diffusion process would effectively require recasting them as diffusion models (e.g., using an FNO/UNO backbone inside a diffusion framework), which is exactly what PRISMA and FunDPS are doing. Since standard FNO/PINO assume dense uniform grids for computing FFT, adapting them to sparse observations requires non-trivial extensions by introducing additional components (e.g., encoders/latent grids), which is beyond the scope of this work.

---

> ### Author Response · Authors · 2025-11-19
> **Author Response to Reviewer H4cn (contd.)**
>
> **Comment 3: More ablation runs for Table 4. Without confidence intervals, the differences in Table 4 are questionable. Some tasks show minimal improvement or even worse performance without PDE residuals. The paper should report mean and std over multiple seeds (code only shows seed:33). Can you report mean and std for +5 random seeds? (Table 4)**
> > Thank you for this suggestion. We have expanded Table 4 to report mean ± std over multiple random seeds (instead of a single seed). Moreover, we have also added results for sparse observations in the same table.
>
> > ### Ablation of Residual Strategies  for Helmholtz & Navier–Stokes (forward/inverse; L2 relative error %)
> | **Model** | **Full Helmholtz (Fwd)** | **Full Helmholtz (Inv)** | **Noisy Helmholtz (Fwd)** | **Noisy Helmholtz (Inv)** | **Full Navier–Stokes (Fwd)** | **Full Navier–Stokes (Inv)** | **Noisy Navier–Stokes (Fwd)** | **Noisy Navier–Stokes (Inv)** |
> |-----------|---------------------------|---------------------------|-----------------------------|-----------------------------|-------------------------------|-------------------------------|--------------------------------|--------------------------------|
> | **PRISMA (w/o PDE res)** | **3.15±0.05%** | 15.67±0.34% | **29.0±0.20%** | 111.87±2.12% | 0.86±0.06% | 7.14±0.16% | 23.04±0.68% | 96.7±1.42% |
> | **PRISMA (PDE res with concat)** | 28.7±0.9% | 51.55±5.75% | 34.5±1.9% | 97.09±1.32% | 0.855±0.005% | 6.77±0.07% | 24.17±0.30% | 86.91±1.55% |
> | **PRISMA (PDE res with SRA) — *ours*** | 3.34±0.14% | **12.47±0.11%** | 30.35±0.45% | **91.85±1.35%** | **0.745±0.055%** | **6.28±0.16%** | **22.2±0.20%** | **80.11±0.31%** |
>
> > ### Ablation on Sparse-Observation Settings  for Helmholtz and Navier–Stokes (L2 relative error %)
> | **Model**                                | **Sparse Helmholtz (Fwd)** | **Sparse Helmholtz (Inv)** | **Sparse Navier–Stokes (Fwd)** | **Sparse Navier–Stokes (Inv)** |
> |------------------------------------------|------------------------------|------------------------------|----------------------------------|----------------------------------|
> | **PRISMA (w/o PDE res)**                 | 36.2±0.15%                   | 66.75±0.19%                  | 29.14±0.18%                      | 31.29±0.20%                      |
> | **PRISMA (PDE res with concat)**         | 94.2±0.50%                   | 124.7±0.24%                  | 31.76±0.14%                      | 33.15±0.34%                      |
> | **PRISMA (PDE res with SRA) — *ours***   | **30.33±0.12%**              | **60.96±0.09%**              | **27.7±0.08%**                   | **29.69±0.01%**                  |
>
> **Comment 4:Claim of multi-scale guidance is somewhat misleading. The paper claims to provide "multi-scale guidance at every layer of UNO", but the implementation uses interpolation by downsampling to the same resolution to match the feature map spatial dimensions. Unlike hierarchical feature extractors, this method appears to not learn different representations of PDE residuals at different scales (simple resize of the same signal). This means the multi-scale is rather a multi-resolution.**
>
> > Thank you for clarifying this. We will revise the paper to use the term “multi-resolution” in place of “multi-scale” accordingly.
>
> **Comment 5:The caption of Figure 4 appears to be incorrect. Caption does not match the figure content (mentions "inference time vs accuracy", but this is not shown)**
>
> > Thank you for bringing this to our attention. This has now been fixed.
>
> **Comment 6: Is there any intuition why FUNDPS outperforms the other methods in the sparse observation case? (Table 7)**
> > Here is a possible explanation for why FunDPS performs the best on sparse observations. FunDPS performs per-sample test time optimization via DPS guidance, using its Tweedie-formula based denoiser, to iteratively refine each sample with hundreds (200-2000) of gradient steps. This allows the model to tightly fit the few observed points, which can reduce error under high sparsity, though at the cost of higher inference time and sensitivity to guidance weights. We view this to be complementary to the strengths of PRISMA. While FunDPS can push accuracy a bit higher when data are very sparse because it fine-tunes each example at test time, PRISMA is built for fast scalable inference without extra test-time optimization.

---

> ### Author Response · Authors · 2025-11-19
> **Author Response to Reviewer H4cn (contd. 2)**
>
> **Comment 7: Did you also try out to integrate magnitude alignment into the attention? Phase alignment and using w_gain appears to perform well, I wonder if magnitude information could be used to omit the w_gain term. Have you ablated using magnitude-only vs. phase-only alignment in frequency domain?**
>
> > Thank you for these suggestions. We have added a new ablation table (see below) that isolates the importance of magnitude-only, phase-only, and gating ablations. In summary, magnitude-only designs, such as the one replacing attention with residual magnitude or using magnitude instead of the phase, consistently underperform when compared with full SRA (our work). We have added this table and its discussion to Appendix E.4.
>
> > ### SRA Ablations
> | **Model**                                   | **Steps (N)** | **Resolution** | **Full Helmholtz (Fwd)** | **Full Helmholtz (Inv)** | **Noisy Helmholtz (Fwd)** | **Noisy Helmholtz (Inv)** | **Full Navier–Stokes (Fwd)** | **Full Navier–Stokes (Inv)** | **Noisy Navier–Stokes (Fwd)** | **Noisy Navier–Stokes (Inv)** |
> |---------------------------------------------|---------------|----------------|---------------------------|---------------------------|-----------------------------|-----------------------------|-------------------------------|-------------------------------|--------------------------------|--------------------------------|
> | **No Gating $(g_{\text{res}}{=}1)$**                        | 20            | 64             | 3.8                       | 28.91                     | 33.8                        | 97.5                        | 1.25                          | 8.2                           | 28.53                          | 86.06                          |
> | **Multiplicative gate(no skip)**                     | 20            | 64             | 3.3                       | 12.98                     | **24.7**                        | 115.07                      | 0.78                          | 6.9                           |**20.7**                           | 92.9                           |
> | **Only mag of PDE residual (no attention)** | 20            | 64             | 3.29                      | 13.42                     | 25.44                       | 115.26                      | 0.8                           | 6.35                          | 24.28                          | 85.72                          |
> | **Attention w/o phase**              | 20            | 64             | 6.5                       | 24.77                     | 34                          | 102.19                      | 0.8                           | 6.54                          | 22.87                          | 92.95                          |
> | **Without guided PDE residual**             | 20            | 64             | 3.52                      | 14.26                     | 31.26                       | 98.99                       | 0.9                           | 6.35                          | 22.9                           | 83.81                          |
> | **Ours (Full SRA)**                      | 20            | 64             | 3.48                      | **12.58**                     | 30.8                        |**93.2**                        | **0.08**                      | **6.12**                          | 22.86                          | **80.4**                        |

---

> ### Author Response · Authors · 2025-11-19
> **Author Response to Reviewer H4cn (contd. 3)**
>
> **Comment 8: Why is complex-valued attention necessary, compared to for example attention in the spatial path?**
>
> > There are two key intuitions behind our use of complex-valued attention in the spectral domain as opposed to attention in the spatial path. First, spectral methods are great at capturing both local and global signals, which is one of the reasons for using UNO as a backbone in PRISMA. To attend to PDE residuals, we need to translate local (and often subtle) PDE violations globally to the entire solution field: a task where spectral methods are a natural fit. In contrast, spatial operations are susceptible to blurring or smoothening local fluctuations in PDE residuals. Second, spectral attention allows precise correction of the solution field based on frequency mode-wise control of PDE residuals. Specifically, we can amplify or suppress the importance of different frequencies of the PDE residual by operating in the spectral space. This is especially important because PDE residual fields are known to show varying sensitivities to different frequencies during loss optimization (see previous works relating failure modes of PINNs to high frequency components of PDE residuals: [1][2]).
>
> > We empirically demonstrate the value of using residuals in the spectral domain as opposed to concatenating residuals in the spatial path in Table 4. We also provide new results by ablating different components of our SRA block in Table 13 of Appendix, to further analyze SRA’s design and isolate the effects of the gate, phase awareness, and skip vs. multiplicative gating.
>
> > [1] Arka Daw, Jie Bu, Sifan Wang, Paris Perdikaris, and Anuj Karpatne. 2023. Mitigating propagation failures in physics-informed neural networks using retain-resample-release (r3) sampling. In Proceedings of the 40th International Conference on Machine Learning (ICML'23), Vol. 202. JMLR.org, Article 288, 7264–7302.
>
> > [2] Krishnapriyan, Aditi, Amir Gholami, Shandian Zhe, Robert Kirby, and Michael W. Mahoney. "Characterizing possible failure modes in physics-informed neural networks." Advances in neural information processing systems 34 (2021): 26548-26560.

---

> > ### Comment · Reviewer_H4cn · 2025-11-25
> >
> > Thanks to the authors for the response.
> >
> > The updated ablation results (Table 4) show that for several tasks, the difference between PRISMA w/o PDE residuals and PRISMA is relatively small. The addition of the SRA block might be application-dependent, and needs further investigation to justify the added complexity.
> >
> > Regarding the baselines, it is now confirmed that FNO/PINO outperform PRISMA on clean data in both accuracy and speed. However, the refusal to compare against FNO/PINO trained with noise augmentation remains a critical flaw. The argument for OOD testing creates a strawman; in practice, one would simply retrain the significantly faster baseline (FNO) on noisy data. Without this comparison, the claims of SOTA performance on noisy observations are unproven.
> >
> > With this in mind and getting informed by the other reviewers perspective/concerns, I cannot justify raising my score.
> > My rating of the `Soundness` for this work remains low, and I advise the authors to perform major revisions in how the paper is written before future submission.

---

> > > ### Author Response · Authors · 2025-12-03
> > >
> > > Thank you for reading our rebuttal. Here are our individual responses to the follow-up comments.
> > >
> > > **Comment: The updated ablation results (Table 4) show that for several tasks, the difference between PRISMA w/o PDE residuals and PRISMA is relatively small. The addition of the SRA block might be application-dependent, and needs further investigation to justify the added complexity.**
> > > >We agree that residuals (or physics guidance) may not always be important, for example in full observation settings, or in easier regimes (test distribution is similar to train). However, physics guidance is especially important when the amount of supervision contained in data alone is not sufficient, for example in noisy, sparse, or inverse settings involving out-of-distribution data. We show that in such cases, using physics guidance in PRISMA results in consistent gains in accuracy with a very small number of steps (20–50). This indicates that PRISMA is able to effectively use physics guidance whenever it is necessary to do so. We have included this clarification in the main paper as well.

---

> > > > ### Author Response · Authors · 2025-12-03
> > > >
> > > > **Comment: Regarding the baselines, it is now confirmed that FNO/PINO outperform PRISMA on clean data in both accuracy and speed. However, the refusal to compare against FNO/PINO trained with noise augmentation remains a critical flaw. The argument for OOD testing creates a strawman; in practice, one would simply retrain the significantly faster baseline (FNO) on noisy data. Without this comparison, the claims of SOTA performance on noisy observations are unproven.**
> > > > > We agree that PINO/FNO are highly competitive on fully observed data regime and even outperform previous diffusion-based methods such as DiffusionPDE and FunDPS in Table 6. However, **PRISMA still has the best average rank across all datasets even in the full observation setting**. Note though that testing on clean data with full observations is not the primary use-case of diffusion-based PDE solvers. Instead, a driving goal of PRISMA is to perform well even on noisy or sparse settings where FNO/PINO are not naturally suited.
> > > >
> > > > > As requested by the reviewer, we have now retrained FNO and PINO with observation-noise augmentation (randomly adding unit gaussian noise to half of the batch with a uniformly sampled noisy-pixel rate) and evaluated on the noisy-observation setting. In the table below, we observe that noise augmentation improves forward errors for both baselines, but inverse errors remain unstable. In contrast, PRISMA achieves the best average rank with 50 steps across Darcy, Poisson, and Helmholtz, without any noise augmentation and maintaining low inference time.
> > > >
> > > > >We want to further emphasize that training on a single noise model (unit Gaussian) can limit generalizability for other noise distributions. In realistic scenarios, sensor noise types and magnitudes vary. For fairness, our main comparisons therefore train all methods without noise augmentation and treat noisy inputs as a common out-of-distribution evaluation. Under this more difficult protocol, PRISMA is consistently robust without tailoring to a specific noise pattern.
> > > >
> > > > >### Comparison of Models under 100% Pixel-Wise Gaussian Noise
> > > > *(L2 relative error for all tasks; error rate for Darcy inverse)*
> > > > | **Model**            | **Steps** | **Infer Time (s)** | **Darcy Fwd** | **Darcy Inv** | **Poisson Fwd** | **Poisson Inv**     | **Helm Fwd** | **Helm Inv**         | **Avg Rank (↓)** |
> > > > |----------------------|-----------|---------------------|----------------|----------------|------------------|-----------------------|---------------|------------------------|------------------|
> > > > | **FNO**              | —         | —                   | 15.70%         | 52.3%          | 25.05%           | 3.0e7%               | 264.8%        | 6.1e5%                | 6.50             |
> > > > | **FNO–Noise Aug**    | —         | —                   | 13.21%       | 49.79%         | 19.78%         | 1.079e7%             | 18.41%      | 6.35e5%              | 4.83             |
> > > > | **PINO**             | —         | —                   | 190.40%        | 52.3%          | 26.64%           | 2.3e6%               | 449.1%        | 1.7e6%                | 6.83             |
> > > > | **PINO–Noise Aug**   | —         | —                   | 36.71%       | 49.78%         | 20.01%         | 8.418e6%             | 19.23%      | 3.14e5%              | 5.17             |
> > > > | **DiffusionPDE**     | 2000      | 213.0               | 49.18%         | 70.08%         | 44.44%           | 130.01%              | 30.97%        | 119.52%               | 5.67             |
> > > > | **FunDPS–200**       | 200       | 4.72                | 26.9%          | *49.39%*       | 78.36%           | *1491.36%*           | *54.99%*      | *629.43%*             | 4.83             |
> > > > | **FunDPS–500**       | 500       | 11.8                | 55.09%         | 49.62%         | 120.0%           | 1772.23%             | 40.31%        | 695.49%               | 5.33             |
> > > > | *PRISMA–20*          | 20        | **0.18**            | *12.29%*       | *23.15%*       | *18.58%*         | *41.75%*             | *17.93%*      | *68.87%*              | *2.00*           |
> > > > | **PRISMA–50**        | 50        | *0.8*               | **12.28%**     | **23.14%**     | **18.35%**       | **41.14%**           | **16.84%**    | **67.93%**            | **1.00**         |

---

### Official Review · Reviewer_DsDE · 2025-10-31

**Soundness:** 1
**Presentation:** 3
**Contribution:** 2
**Rating:** 2
**Confidence:** 4

**Summary:**

The paper introduces PRISMA (PDE Residual Informed Spectral Modulation with Attention), a novel conditional diffusion neural operator for solving forward and inverse problems in the context of PDEs. To avoid the cost and potential instabilities of inference-time guidance, they propose to condition the operator directly on the (noisy and masked) PDE residuals using a spectral residual attention block. Similar to existing solvers, PRISMA operates on a joint space of observations and parameters to enable the solution of forward and inverse problems with sparse and noisy observations. Due to the guidance-free inference, PRISMA can outperform existing solvers in terms of accuracy vs. inference cost.

**Strengths:**

1. The spectral residual attention block seems to be a novel module for conditioning diffusion models/operators on residuals from sparse and noise observations. This can lead to improved performance, in particular for noisy observations (potentially due to the learned gating mechanism). Since no guidance is needed during inference, this further reduces the required number of steps and inference time compared to DiffusionPDE and FunDPS.
2. The paper includes a comprehensive set of experiments (from DiffusionPDE) and provides ablation studies on the conditioning type and number of required diffusion steps (showing that performance saturates quickly at 20-50 steps and validating the use of a low number of steps). Moreover, it is shown that PRISMA can improve statistics of the PDE residuals over the considered two baselines.

**Weaknesses:**

1. A major concern is that PRISMA considers a different setting than the two main baselines. The baselines only require paired data, but no prior knowledge of the PDE or the corruptions (type of masking/noise/etc.) during training. In other words, these guidance-based methods are *agnostic* to the corruptions and just rely on inference-time control. On the other hand, PRISMA assumes knowledge of the type of masks, noise, and PDE equations *during training*. This is a more restrictive setting and (unsurprisingly) also leads to better performance.
2. The framework of jointly modeling observations and parameters (and thus being able to solve forward and inverse problems) is taken directly from DiffusionPDE (finite-dim.) and FunDPS (infinite-dim.).
3. As mentioned in the paper, the performance is only evaluated on the dataset and baselines provided by DiffusionPDE which do not consider practically relevant problems on full time-intervals (instead of initial/terminal time problems). Due to its dependence on FFT, it seems also non-trivial to extend PRISMA to problems on irregular geometries.
4. Some of the claims seem to be too strong:
    - The proposed guidance is still not “pointwise” since the Fourier transform has a global dependency.
    - The method is not fully "gradient-free" during inference, since the gradients appearing in the PDE still need to computed (e.g. using finite-differences or Fourier differentiation).
    - While the methods has a better performance vs. inference cost trade-off, the statement that it is “at-par or better in accuracy” is not true given the sparse-observation results reported in Table 7.

**Minor:**

- The caption of Figure 6 seems to be wrong (and coincides with the caption of Figure 4).

**Questions:**

- It would be interesting to see the ablation when directly conditioning on the sparse and noisy observations instead of the PDE residual.
- It seems that the guidance weight for DiffusionPDE/FunDPS is not tuned properly since the reported performance is sometimes decreasing when increasing the number of steps.
- How much slower is the training due to the computations of residuals and the additional spectral residual attention block?
- Does PRISMA train different models for different noise levels for the (sparse) observation (and if not, how is the noise level sampled during training). Moreover, how are the masks sampled during training?
- It would be interesting to ablate the effect of the gating mechanism in the noisy setting.

---

> ### Author Response · Authors · 2025-11-19
> **Author Response to Reviewer DsDE**
>
> Thank you for the comments. Please find our responses to individual comments below.
>
> **Comment 1: A major concern is that PRISMA considers a different setting than the two main baselines. The baselines only require paired data, but no prior knowledge of the PDE or the corruptions (type of masking/noise/etc.) during training. In other words, these guidance-based methods are agnostic to the corruptions and just rely on inference-time control. On the other hand, PRISMA assumes knowledge of the type of masks, noise, and PDE equations during training.   This is a more restrictive setting and (unsurprisingly) also leads to better performance. Does PRISMA train different models for different noise levels for the (sparse) observation (and if not, how is the noise level sampled during training). Moreover, how are the masks sampled during training?**
>
> > We would like to clarify a few important points about the training of PRISMA:
> > 1. **PRISMA training is noise-free.** We do not train PRISMA by augmenting inputs or outputs with noise corruptions. We only use noisy inputs during inference, making our training process noise-agnostic same as baseline methods.
> >  2. **PRISMA training is agnostic to sparsity levels seen during inference.** We train PRISMA on a unified set of diverse sparsity levels in inputs (sampled from a Uniform distribution as detailed below) rather than a fixed sparsity level or missingness pattern. More crucially, we do not assume knowledge of or tune our model to the type of masks or sparsity levels encountered during inference, which still represent unseen testing scenarios just like baseline methods.
> > Specifically, we use the following sampling procedure for creating binary masks ($M_a$ or $M_u$) in the training of PRISMA.
>  	- For every batch, we introduce sparse masks for 50% of samples while the remaining samples have no sparsity (i.e., full observation).
>     - For every sparse mask, we sample its observation rate $ p_{\text{obs}}$ on-the-fly using the following equation favoring high-sparsity scenarios ($p_{\text{obs}} \approx 1$) that are more challenging while still covering low-sparsity scenarios:
> >  \\[
> p\_{\text{obs}} \sim
> \begin{cases}
> \text{Uniform}\\Big[\\text{min} \, \\text{min} + 0.1(\\text{max}-\\text{min})\\Big], & \\text{with probability } 0.5, \\\\[8pt]
> \\text{min} + \\left(1 - U^{\\alpha}\\right)(\\text{max}-\\text{min}), & \\text{with probability } 0.5, U\sim\\text{Uniform}(0,1).
> \end{cases}
> >\\]
> >where \\(\\alpha = 3 , min=0.01\\)  and \\(max=0.5.\\)
> >Note that $p_{\text{obs}}$ varies between 0.01 (99\% sparse) and 0.5 (50\% sparse). We then sample from an i.i.d. Bernoulli $(p_{\text{obs}})$ distribution over grid locations (uniform across all spatial positions) to create binary masks. We train a single model using this sampling procedure and once trained, we evaluate it on varying sparsity levels (Table 7 in Appendix shows results with 97% sparsity), noise levels and full observations.
>
> >3. **PRISMA is trained using PDE knowledge.** One of the most important differences between our work and baselines is that we push the optimization of PDE loss to training, so that inference is lightweight and gradient descent-free. We agree that this makes our training PDE-informed, while the training of baselines is PDE-agnostic. However, this does not make our model any more restrictive, as the knowledge of PDE is an invariant property of the system that we do not expect to change between training and inference. Moreover, baseline methods (DiffusionPDE and FunDPS) are allowed to perform per-sample optimization of PDE loss during inference, allowing them to control their generated outputs in accordance with PDE guidance. This is in contrast to PRISMA, where we move this dependence to training (via residual-aware conditioning) so that inference is feed-forward (no guidance, no per-sample optimization).

---

> ### Author Response · Authors · 2025-11-19
> **Author Response to Reviewer DsDE (contd.)**
>
> **Comment 2: The framework of jointly modeling observations and parameters (and thus being able to solve forward and inverse problems) is taken directly from DiffusionPDE (finite-dim.) and FunDPS (infinite-dim.).**
> > We agree that the idea of jointly modeling observations and parameters using a unified framework is similar to DiffusionPDE and FunDPS. However, there are a few key differences between our framework and that of baselines making our work novel.
> > 1. **Conditional vs. Unconditional Modeling.** While previous works consider an unconditional model during training and inference (with test-time optimization) to unify forward and inverse problems, PRISMA uses a *conditional* unified model. This allows us to use input observations (and masks for forward/inverse) as conditions during inference, making our inference purely feed-forward and eliminating per-sample optimization that results in 50-250x speedup.
> > 2. **Use of SRA Block.** While PRISMA uses a similar UNO backbone in the Fourier space as FunDPS, the main novelty in our framework is the use of Spectral Residual Attention (SRA) blocks that are PDE residual-aware, which is absent in both baselines. By using SRA blocks, we avoid backpropagating PDE loss during inference, which is a primary contribution of our work.
>
> **Comment 3: As mentioned in the paper, the performance is only evaluated on the dataset and baselines provided by DiffusionPDE which do not consider practically relevant problems on full time-intervals (instead of initial/terminal time problems). Due to its dependence on FFT, it seems also non-trivial to extend PRISMA to problems on irregular geometries.**
>
> > We agree that applying our work on more challenging problems is a valuable contribution. However, we chose the exact same suite of benchmark PDE experiments as used in recent works (DiffusionPDE and FunDPS) to maintain consistency and fairness in comparison of results. Note that a practical challenge in evaluating baseline methods such as DiffusionPDE on new datasets is their high sensitivity to hyper-parameter settings and issues with reproducibility that have been documented in previous works. To ensure a fair comparison, we used reported checkpoints of baselines on known datasets without extending them, which can certainly be the scope of future works.
>
> > Regarding extensions on full time intervals, note that PRISMA currently works on single channel inputs and outputs, which naturally aligns it with initial-to-terminal problems.  To show the generalizability of PRISMA to spatio-temporal problems, we performed new experiments comparing PRISMA with DiffusionPDE on the 1D-Burgers equation where the goal is to predict the full-time solution field given sparse spatial observations across all timesteps. Following the experiment setup of DiffusionPDE, we randomly selected 5 out of 128 spatial sensors for creating sparse inputs. We reproduced DiffusionPDE’s results using their publicly released checkpoint and default hyperparameters. The relative errors on this experiment are summarized in the Table below.
>
> >### Relative Error (%) on 1D Burgers equation
> | **Method**       | **Relative Error (%)** | **Steps** |
> |------------------|------------------------|-----------|
> | **DiffusionPDE** | 10.39%                 | 2000      |
> | **PRISMA**       | 9.33%                  | 20        |
>
> >  We can see that PRISMA achieves a slightly better accuracy than DiffusionPDE while reducing inference from 2000 guidance steps to 20 feed-forward steps. We have added more details and visualizations in Appendix F of the paper.
>
> >  PRISMA can also be extended to irregular geometries by adopting a latent-space diffusion formulation where we first encode inputs and outputs to a regular latent grid, learn diffusion neural operators, and decode the latent space back to the original space of inputs and outputs. Since our SRA block is backbone-agnostic, it can also be plugged with any diffusion backbone. We will discuss these points in the future works section.
>
> **Comment 4: The proposed guidance is still not “pointwise” since the Fourier transform has a global dependency.**
>
> > We agree that our SRA module operates in the spectral domain and thus has global support. However, the distinction we would like to make is that while previous works aggregate the PDE residual field into a single number, we use the disaggregated PDE residuals in the spectral domain and use them as keys at every frequency mode to compute its spectral cross-attention with the feature maps in Equation 6. This is what we mean by pointwise guidance, which operates in the spectral domain (and hence is not spatially local) but considers variations in PDE residuals across frequency modes (points in spectral domains) without aggregation.

---

> ### Author Response · Authors · 2025-11-19
> **Author Response to Reviewer DsDE (contd. 2)**
>
> **Comment 5: The method is not fully "gradient-free" during inference, since the gradients appearing in the PDE still need to computed (e.g. using finite-differences or Fourier differentiation).**
> > Thank you for this clarification. Our use of the term “gradient-free inference” can be made tighter as it refers to the absence of reverse-mode autograd of PDE or observation loss for test-time optimization. In other words, while we do compute PDE residuals using finite differences and use it in the forward pass, we do not backpropagate this loss w.r.t. the sample or model parameters. We will clarify this detail and update the phrasing to be “gradient-descent free” instead of “gradient-free.”
>
> **Comment 6: While the method has a better performance vs. inference cost trade-off, the statement that it is “at-par or better in accuracy” is not true given the sparse-observation results reported in Table 7.**
> > We agree that the phrasing “at-par or better in accuracy” is rather strong and there are a few cases with sparse observations in Table 7 where our errors are slightly higher than baselines while still being ~15x to ~65x faster. In the table below, we show that relative to the strongest baseline for sparse observations (FunDPS-500), PRISMA is within small relative % errors while being significantly faster:
> > ### Performance and Latency Comparison
> | **Model (steps)** | **Darcy For Δ** | **Darcy Inv Δ** | **NS For Δ** | **NS Inv Δ** | **Per-sample latency** | **Speedup vs FunDPS-500** |
> |-------------------|------------------|------------------|--------------|--------------|--------------------------|-----------------------------|
> | **PRISMA-20**     | +0.64            | +1.62            | +0.58        | +3.25        | 0.18 s                   | ~65×                        |
> | **PRISMA-50**     | +0.62            | +1.62            | +0.53        | +3.02        | 0.80 s                   | ~15×                        |
>
> >(Δ = relative percentage gap to FunDPS-500; lower is better.)
>
> > We will revise our phrasing to “competitive accuracy at substantially lower inference cost” to reflect this change.
>
> **Comment 7: The caption of Figure 6 seems to be wrong (and coincides with the caption of Figure 4).**
> > Thank you for bringing this to our attention. This has now been fixed.
>
> **Comment 8: It would be interesting to see the ablation when directly conditioning on the sparse and noisy observations instead of the PDE residual.**
> > Thank you for this suggestion. We believe we have already provided results for this ablation in Table 4. Specifically, the first row of Table 4 is for the Observation-only (w/o PDE residual) variant, which uses no residual information at all, and conditions directly on the sparse/noisy observations without any PDE residuals or guidance. We’ve updated Table 4 to report mean ± std over multiple seeds and to report results of these ablations on Sparse observation settings as well. We apologize if we misunderstood this comment and if you had a different ablation in mind, we will be happy to include it.
>
> **Comment 9: It seems that the guidance weight for DiffusionPDE/FunDPS is not tuned properly since the reported performance is sometimes decreasing when increasing the number of steps.**
>
> > We reproduced DiffusionPDE/FunDPS using their exact public configs, released checkpoints, per-equation default guidance weights, and suggested step counts. Given their high sensitivity to seed/config setups (see Appendix D) that is also reported in prior work (see Appendix I of FunDPS paper), we chose not to perform any hyper-parameter tuning of baselines to be faithful in reproducing their original works. We quote prior numbers from their tables wherever we are able to reproduce their results within a reasonable range. Otherwise, we re-ran using their default configurations (on multiple seeds) and reported the observed values (these revised numbers are highlighted using asterisk in Table 6).
>
> > Regarding the observation that the performance of baselines sometimes decreases with more steps: this shows up primarily in the noisy-observation regime, where guidance-based solvers perform per-sample test-time optimization using residuals computed from corrupted observations. Since the residuals are noisy, it is easy to overfit to noise or amplify unstable high-frequency residual components as we increase the number of steps. This is precisely the failure mode that PRISMA is designed to mitigate by using residual guidance at training rather than inference and using a noise-aware gating mechanism $g_{\text{res}}$ that dynamically controls the denoising process based on the relevance of PDE residuals in noisy settings.

---

> ### Author Response · Authors · 2025-11-19
> **Author Response to Reviewer DsDE (contd. 3)**
>
> **Comment 10: How much slower is the training due to the computations of residuals and the additional spectral residual attention block?**
>
> > Table below shows training and inference time comparisons of PRISMA, FunDPS, and DiffusionPDE on a single A100, averaged across batches for Darcy at 128 resolution. We observe that PRISMA’s residual computation in SRA adds ~5.8 sec/iteration over a plain Diffusion-UNO (FunDPS) and is 28% slower. However, PRISMA is still ~2.6× faster when compared with DiffusionPDE (UNet backbone) per training iteration. PRISMA’s inference is much faster at 0.18 s/sample since it does not involve test-time optimization.
>
> >### Training and Inference Speed Comparison (Relative to PRISMA)
> | **Method**       | **Train speed (sec/iteration)** | **Δ vs PRISMA**     | **20k iteration wall-clock (h)** | **Inference (s/sample)** | **Δ vs PRISMA**        | **Inference (1k samples)** |
> |------------------|-----------------------------|-----------------------|------------------------------|---------------------------|--------------------------|-----------------------------|
> | **PRISMA**       | 26.0                        | —                     | 144.4                        | 0.18                      | —                        | ~3 min                      |
> | **FunDPS**       | 20.2                        | 22.3% faster          | 112.2                        | 11.78                     | ≈65.4× slower            | ~3 h 16 m                   |
> | **DiffusionPDE** | 58.73                       | 126% slower           | 326.3                        | 213                       | ≈1,183× slower           | ~59 h 10 m
>
>
>
> **Comment 11: It would be interesting to ablate the effect of the gating mechanism in the noisy setting.**
>
> > We have added a new table (see below) that shows the effect of the gate ablations in full and noisy regimes. We see that removing the gate (No Gating) consistently degrades performance in noisy Helmholtz and noisy Navier–Stokes. A purely multiplicative gate (no skip connection) is less stable and underperforms. The full SRA (phase-aware attention + learned per-mode gains + scalar gate) yields the best results in noisy settings. We have added this table and discussion in Appendix E.4.
>
> > ### SRA Ablations
> | **Model**                                   | **Steps (N)** | **Res** | **Full Helmholtz (Fwd)** | **Full Helmholtz (Inv)** | **Noisy Helmholtz (Fwd)** | **Noisy Helmholtz (Inv)** | **Full Navier–Stokes (Fwd)** | **Full Navier–Stokes (Inv)** | **Noisy Navier–Stokes (Fwd)** | **Noisy Navier–Stokes (Inv)** |
> |---------------------------------------------|---------------|----------------|---------------------------|---------------------------|-----------------------------|-----------------------------|-------------------------------|-------------------------------|--------------------------------|--------------------------------|
> | **No Gating $(g_{\text{res}}{=}1)$**                        | 20            | 64             | 3.8                       | 28.91                     | 33.8                        | 97.5                        | 1.25                          | 8.2                           | 28.53                          | 86.06                          |
> | **Multiplicative gate(no skip)**                     | 20            | 64             | 3.3                       | 12.98                     | **24.7**                        | 115.07                      | 0.78                          | 6.9                           |**20.7**                           | 92.9                           |
> | **Only mag of PDE residual (no attention)** | 20            | 64             | 3.29                      | 13.42                     | 25.44                       | 115.26                      | 0.8                           | 6.35                          | 24.28                          | 85.72                          |
> | **Attention w/o phase**              | 20            | 64             | 6.5                       | 24.77                     | 34                          | 102.19                      | 0.8                           | 6.54                          | 22.87                          | 92.95                          |
> | **Without guided PDE residual**             | 20            | 64             | 3.52                      | 14.26                     | 31.26                       | 98.99                       | 0.9                           | 6.35                          | 22.9                           | 83.81                          |
> | **Ours (SRA)**                      | 20            | 64             | 3.48                      | **12.58**                     | 30.8                        |**93.2**                        | **0.08**                      | **6.12**                          | 22.86                          | **80.4**                        |

---

### Author Response · Authors · 2025-11-22
**Global Response to Review Comments**

We sincerely thank all the reviewers for their constructive and detailed feedback. We are encouraged that the reviewers appreciated several key aspects of our work and found:

1.	Our Spectral Residual Attention (SRA) to be a novel PDE-informed conditioning module for conditioning diffusion models/operators on residuals [*Reviewers DsDE, H4cn, 851Q, hiiL*].

2.	Our guidance-free 20-step inference achieving 15-250× speedups while maintaining competitive accuracy to be a significant contribution [*Reviewers H4cn, hiiL, DsDE*].

3.	Our approach to show robustness under heavy observation noise and to improve residual statistics relative to diffusion baselines in several settings [*Reviewers DsDE, 851Q, hiiL*].

4.	Our conditional operator to unify forward and inverse solving across full, sparse, and noisy observation regimes within a single framework [*Reviewers H4cn, hiiL*].

5. Our empirical evaluation to be comprehensive across five PDE benchmarks, with informative ablations (conditioning type, step counts) [*Reviewers DsDE, H4cn, hiiL*].

6.	Our released code and overall clarity to aid verification and reproducibility [*Reviewers H4cn, 851Q*].

To address the reviewer's comments, we have performed additional experiments and added new figures, tables, and text explanations in the revised manuscript (revisions highlighted in red). **Here is a summary of the new experiments:**

1.	Performed ablations to study the SRA block’s architecture and re-ran Table 4 ablations with a different seed. (Table 4 & Appendix Table 12) (addressing Reviewers DsDE, H4cn, hiiL).
2.	Validated the feasibility of full-trajectory prediction on 1D Burgers for full-time interval prediction. (Appendix Section G) (addressing Reviewers DsDE, hiiL).
3.	Evaluated robustness across noise difficulty levels by corrupting 10%, 30%, 50%, and 90% of pixels for both forward and inverse settings (Appendix Tables 14-17) (addressing Reviewer H4cn).
4.	Performed super-resolution analysis and a compute-cost comparison (training/inference time and parameters) (Appendix Section G) (addressing Reviewer hiiL).

We would like to address some of the shared concerns raised by the reviewers in our general response below.

**Comment 1:  PRSIMA Training**

>1. **PRISMA training is noise-free.** We do not train PRISMA by augmenting inputs or outputs with noise corruptions. We only use noisy inputs during inference, making our training process noise-agnostic same as baseline methods.
>2. **PRISMA training is agnostic to sparsity levels seen during inference.** We train PRISMA on a unified set of diverse sparsity levels in inputs (sampled from a Uniform distribution) rather than a fixed sparsity level or missingness pattern. More crucially, we do not assume knowledge of or tune our model to the type of masks or sparsity levels encountered during inference, which still represent unseen testing scenarios just like baseline methods.
>3. **PRISMA is trained using PDE knowledge.** One of the most important differences between our work and baselines is that we push the optimization of PDE loss to training, so that inference is lightweight and gradient descent-free. We agree that this makes our training PDE-informed, while the training of baselines is PDE-agnostic. However, this does not make our model any more restrictive, as the knowledge of PDE is an invariant property of the system that we do not expect to change between training and inference. Moreover, baseline methods (DiffusionPDE and FunDPS) are allowed to perform per-sample optimization of PDE loss during inference, allowing them to control their generated outputs in accordance with PDE guidance. This is in contrast to PRISMA, where we move this dependence to training (via residual-aware conditioning) so that inference is feed-forward (no guidance, no per-sample optimization).
Once trained, a single model is used to evaluate across multiple settings: full, noisy (different % of pixel corruption, or noise $\sigma$, sparse (different sparsity levels)

**Comment 2: Accuracy-Latency Tradeoff**
>PRISMA offers a superior accuracy-latency trade-off. Under full observations it is competitive across all PDEs and attains the second-best average rank while using only 20–50 steps (no per-instance test-time optimization). In sparse regimes, PRISMA’s errors are within small relative % gaps of the strongest baseline (FunDPS-500) yet it is ~15x – 65x faster per sample. In noisy settings, PRISMA is state-of-the-art across both noise scales ($\sigma$) and %-pixel corruption levels, without sacrificing speed. Concretely, for 1,000 samples, PRISMA completes inference in ~3 min, vs. FunDPS-500 needs ~3.5 hours and DiffusionPDE is at ~60 hours. This speedup is enabled by moving residual guidance to training (via SRA) and keeping feed-forward inference, yielding a single unified model that handles full/partial/noisy observations at substantially lower inference cost while maintaining competitive accuracy.

---

> ### Author Response · Authors · 2025-11-22
> **Continued Global Response**
>
> We hope that our responses to the individual review comments provided below address the main concerns of the reviewers. If we missed any details, we will be happy to provide more clarifications during the discussion period. If our responses have adequately addressed your concerns, we kindly request that you consider revising your scores. Thank you very much for your time and effort.

---

### Author Response · Authors · 2025-12-03
**Summary Response to Area Chairs**

Thank you for the opportunity to summarize our discussion with the reviewers so far to the Area Chair.

**Common strengths identified by reviewers:**
>We are grateful that the reviewers appreciated the novelty of our work *(Reviewers DsDE, H4cn, 851Q, hiiL)* in developing a diffusion-based PDE solver that delivers ~15–250× speedups while having competitive accuracy *(Reviewers H4cn, hiiL, DsDE).* Reviewers also noted our robustness under heavy observation noise *(Reviewers DsDE, hiiL)* and the fact that we can jointly handle both forward and inverse tasks across full, sparse, and noisy regimes *(Reviewers H4cn, hiiL).*

**Clarifying our main contribution:**
>PRISMA is the first single unified model that solves full, partial and noisy cases *without per-instance test-time optimization*, pushing the accuracy-latency frontier across all regimes rather than chasing peak accuracy in a narrow regime. **PRISMA samples in just 20–50 steps and is able to process 1,000 samples in ~3 minutes. In contrast, FunDPS needs ~3 hours and DiffusionPDE needs ~69 h!** Under noisy observations, PRISMA achieves the best average rank while retaining this speed, and on full and sparse settings it remains competitive (best/second-best rank), *yet is orders of magnitude faster!* We also show results on super-resolution experiments and a spatio-temporal extension on 1D Burgers equation.

**Clarifying reviewer concerns:**
>We recognize that some of the low ratings of the reviewers stem from misunderstandings about our methodology that we have fully addressed with new experiments and clarifications summarized below.

>1.	**Clarification on noise augmentation:**
Some reviewers have misunderstood that diffusion-based methods including our work were trained with input noise augmentation, which they were not. All models (PRISMA, FNO/PINO, DiffusionPDE, FunDPS) were trained without adding noise to PDE observations. The stochastic noise used in diffusion training is different and should not be confused with input noise augmentation. **We only use noisy inputs during testing to demonstrate out-of-distribution generalization.** We also show in our response to Reviewer H4cn that even after retraining baselines such as FNO/PINO with input-noise augmentation, they still do not achieve similar performance as PRISMA especially on inverse problems.

>2.	**Clarification on performance comparisons:**
We acknowledge that across the broad suite of experiments considered in this work (involving full, noisy, and sparse settings on four benchmark PDEs for both forward and inverse problems), there are a few scenarios where PRISMA does not achieve the absolute best performance. However, **PRISMA consistently attains the highest overall rank across all experiments and even in cases where it is not the top-performing method, it remains highly competitive while being orders of magnitude faster.** We summarize performance comparisons of PRISMA in the three settings as follows.
-	**Noisy observations.** PRISMA attains the best average rank in this more complex setting and achieves accuracy values unmatched by any other baseline, while maintaining low inference times.
-	**Full observations.** While FNO/PINO show strong performance in this setting with better average ranks even compared to DiffusionPDE and FunDPS, PRISMA still achieves the best average rank across all PDEs while using only 20–50 steps.
-	**Sparse observations.** While FunDPS is accurate in some sparse cases (which is the primary use-case that FunDPS was designed for), PRISMA is still ~15×–65× faster per sample than FunDPS-500, with small accuracy gaps (second-best average rank).


>3.	**Clarification on Navier–Stokes residual computation:**
We sincerely appreciate Reviewer 851q’s feedback in identifying an issue in DiffusionPDE’s codebase in the computation of Navier-Stokes PDE residuals that has been inadvertently propagating in the diffusion-based PDE community. We have revised all our results on the Navier-Stokes problem after fixing this issue. We show that PRISMA still remains competitive at 20–50 steps, achieving the best average rank (1.33) on Navier-Stokes using ~100× fewer steps than DiffusionPDE (2000).

>4.	**Extending to spatio-temporal & irregular meshes:**
To show generalization on spatio-temporal problems, we have added new results on a full trajectory 1D Burgers experiment, where PRISMA achieves 9.33% at 20 steps versus DiffusionPDE’s 10.39% at 2000 steps. For more general settings, we outline two extensions as part of future work: (1) incorporating stacked time slices within the same spectral backbone for spatio-temporal problems, and (2) using latent-space diffusion models for irregular geometries. Our SRA module is backbone-agnostic and compatible with these extensions.

---

> ### Author Response · Authors · 2025-12-03
> **Summary Response to Area Chairs (contd.)**
>
> Below is a summary of the new experiments we have performed, and the revisions made to the manuscript, to address the reviewer's comments:
> >1. Performed ablations to study the SRA block’s architecture and re-ran Table 4 ablations with a different seed. *(Table 4 & Appendix Table 12) (addressing Reviewers DsDE, H4cn, hiiL).*
> >2. Validated the feasibility of full-trajectory prediction on 1D Burgers for full-time interval prediction. *(Appendix Section G) (addressing Reviewers DsDE, hiiL).*
> >3. Evaluated robustness across noise difficulty levels by corrupting 10%, 30%, 50%, and 90% of pixels for both forward and inverse settings *(Appendix Tables 14-17) (addressing Reviewer H4cn).*
> 4. Performed super-resolution analysis and a compute-cost comparison (training/inference time and parameters) *(Appendix Section G) (addressing Reviewer hiiL).*
> 6. Performed noise augmentation experiment by retraining FNO and PINO with input-noise augmentation. *(addressing Reviewer H4cn).*
> 7. Corrected the Navier–Stokes residual computation by using the vorticity-transport (momentum) residual and switching to next-step prediction with Δt = 0.1. All tables reflect the updated numbers and Appendix C has been revised to detail the new residual computation.  *(Table 3, Appendix Tables 6&7, Section C) (addressing Reviewer 851Q).*
> 8. Made minor clarifications regarding terminology throughout the paper such as “gradient-free” to “gradient descent-free” and “at-par or better” to “competitive accuracy,” and “multi-scale” to “multi-resolution.” Finally, “100% Gaussian noise” means unit Gaussian added at every pixel. *(addressing Reviewer DsDE, H4cn, 851Q).*

---

### Meta-Review · Area_Chair_GXeN · 2026-01-05

**Summary:**

The paper introduces a conditional diffusion neural operator for solving partial differential equations (PDEs). Its main contribution lies in directly incorporating PDE residuals into the operator layers, rather than enforcing them through an additional residual loss during training or inference, as done in prior diffusion-based approaches. This is achieved via a conditional mechanism implemented through residual attention blocks operating in the spectral domain. The proposed model is trained on the joint space of parameters and observations, enabling it to address both forward and inverse problems, as well as to operate under sparse and noisy observation settings. Experimental validation is conducted on several classical benchmark datasets.

The reviewers acknowledge the novelty of the proposed conditional mechanism for integrating PDE residuals and agree that the method yields significant speedups compared to alternative diffusion-based models. However, they also raise several concerns. A major issue is the fairness of the experimental comparison: while the proposed framework explicitly leverages prior knowledge of the governing PDE, most baselines are purely data-driven, making direct comparisons questionable. Another concern is that, when operating on fully observed data, the proposed method underperforms some baselines in both speed and accuracy, despite relying on additional PDE information. Moreover, the framework is evaluated primarily on single-step prediction at a final time, rather than on full trajectory forecasting. One reviewer also points out that the experimental protocol used for some datasets—borrowed from prior work—is flawed; although this issue originates from the referenced methodology rather than the authors’ implementation, it nonetheless casts doubt on the reported results.

**Reviewer Concerns:**

The authors provided extensive responses to the reviewers’ comments and added new experiments and ablation studies which makes their contribution stronger. While these efforts address several points, they do not fully resolve all concerns raised during the review process. Overall, the work appears promising, but a substantial revision would be necessary before resubmission.

**Reviewer Scores:**

RDsDE, rating 2, the authors provided a very strong rebuttal with several additional experiments that solve several concerns raised by the reviewer, should probably raise their score

RH4cn, rating 4, indicates that considering the rebuttal and other reviewer concerns they will not increase their score.

R851Q, rating 2, not convinced by the rebuttal for some experimental results, would probably not increase their score.

RhiiL, rating 4, extensive answer with new results, might increase their score.

---

### Decision · Program_Chairs · 2026-01-26

Reject